# Proof of concept for a new sensor to monitor marine litter from space

Andrés Cózar ®[1,13] ✉, Manuel Arias ®[2,3,4,13] ✉, Giuseppe Suaria ®[5], Josué Viejo[1], Stefano Aliani[5], Aristeidis Koutroulis ®[6], James Delaney[7], Guillaume Bonnery[8], Diego Macías[9], Robin de Vries ®[10], Romain Sumerot[11], Carmen Morales-Caselles ®[1], Antonio Turiel[2], Daniel González-Fernández ®[1] & Paolo Corradi[12]

Worldwide, governments are implementing strategies to combat marine litter. However, their effectiveness is largely unknown because we lack tools to systematically monitor marine litter over broad spatio-temporal scales. Metre-sized aggregations of floating debris generated by sea-surface convergence lines have been reported as a reliable target for detection from satellites. Yet, the usefulness of such ephemeral, scattered aggregations as proxy for sustained, large-scale monitoring of marine litter remains an open question for a dedicated Earth-Observation mission. Here, we track this proxy over a series of 300,000 satellite images of the entire Mediterranean Sea. The proxy is mainly related to recent inputs from land-based litter sources. Despite the limitations of in-orbit technology, satellite detections are sufficient to map hot-spots and capture trends, providing an unprecedented source-to-sink view of the marine litter phenomenon. Torrential rains largely control marine litter inputs, while coastal boundary currents and wind-driven surface sweep arise as key drivers for its distribution over the ocean. Satellite-based monitoring proves to be a real game changer for marine litter research and management. Furthermore, the development of an ad-hoc sensor can lower the minimum detectable concentration by one order of magnitude, ensuring operational monitoring, at least for seasonal-to-interannual variability in the mesoscale.

Marine litter (ML) mirrors the failures of our civilization to manage waste[1,2]. In a few decades, ML has grown into a global environmental threat, recognised as an urgent priority on international sustainability agendas[3,4]. Governments and businesses around the world are rolling out a plethora of measures to combat ML. However, field surveys are proving insufficient to track ML trends on a global scale, and the effectiveness of the action plans against ML remains uncertain[5].

ML is any human-made item that ends up in marine environments. The diversity is huge, but about 80% of these items are made of plastic (95% in the ocean surface waters)[6], making this material the main concern for managers[4] and the focus of monitoring[5]. The first global maps of

[1]Departamento de Biología, Facultad de Ciencias del Mar y Ambientales, Universidad de Cádiz and European University of the Seas (SEA-EU), Puerto Real, Spain. [2]Institute of Marine Sciences (ICM-CSIC), Barcelona Expert Center, Barcelona, Spain. [3]ARGANS France, Sophia-Antipolis, cedex, France. [4]Universitat Politècnica de Catalunya (UPC), Barcelona, Spain. [5]Istituto di Scienze Marine - Consiglio Nazionale delle Ricerche (ISMAR-CNR), Lerici, La Spezia, Italy. [6]Technical University of Crete, School of Chemical and Environmental Engineering, Chania, Greece. [7]ARGANS Ltd., Plymouth, United Kingdom. [8]Airbus Defence and Space, Toulouse, France. [9]European Commission, Joint Research Centre, Ispra, Italy. [10]The Ocean Cleanup, JH Rotterdam, The Netherlands. [11]ACRI-ST, Sophia-Antipolis, France. [12]European Space Agency - ESTEC, Noordwijk, The Netherlands. [13]These authors contributed equally: Andrés Cózar, Manuel Arias. ✉e-mail: andres.cozar@uca.es; marias@icm.csic.es

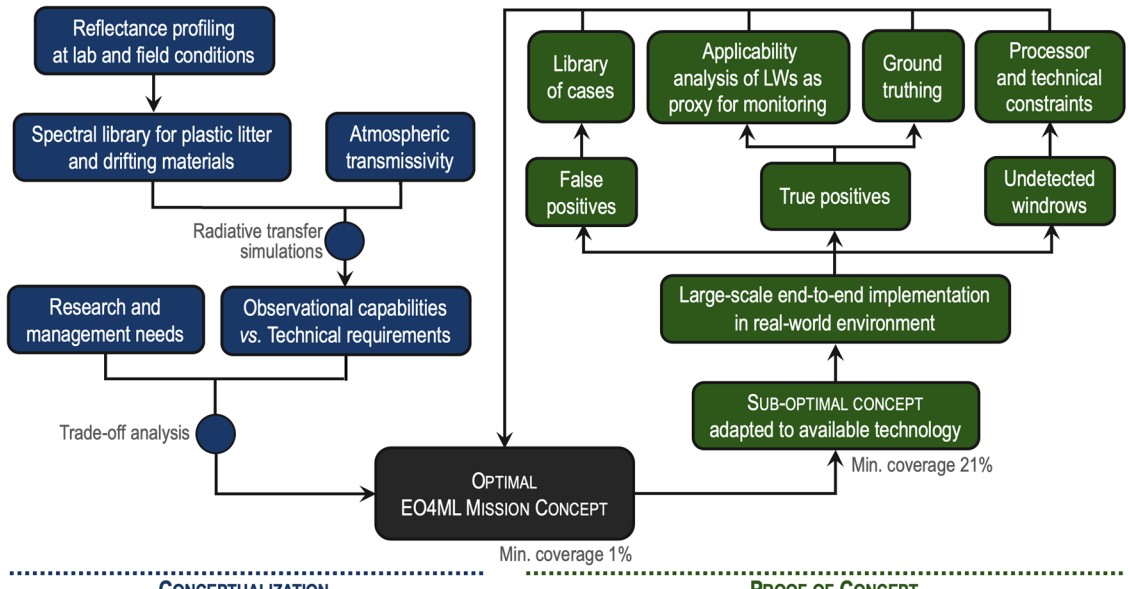

**Fig. 1 | Roadmap followed for the definition of the optimal mission concept (blue) and for the proof of concept (green).** Min. coverage refers to the minimum fraction of plastic surface coverage required to identify a pixel with a true positive, estimated for both optimal mission and the proof of concept. An extended version of the workflow can be found in Fig. S1.

plastic debris became available for the ocean surface a decade ago[7,8]. Continued efforts to extend data coverage have produced important results[9], but field measurements remain sparse and unexplored areas still cover most of the ocean. In this framework, satellite-based observations are one of the best hopes for large-scale monitoring of ML[10]. Yet, the fraction of ocean surface physically covered by plastic in the most polluted regions is estimated to be below 0.02%, from 200 to 0.1 parts per million[11,12]. The weakness of the radiative signal reflected by the main component of ML imposes strong constraints for its direct monitoring from space, at least following the standard approaches used for other ocean constituents such as phytoplankton or suspended sediments.

An alternative opportunity to monitor ML from space lies in the use of proxies, like the metre-sized aggregations of floating debris, the so-called surface slicks, streaks, filaments or litter windrows (LWs)[13–15]. The action of wind, tides, river plumes and ocean density currents often generate surface convergence zones in the sub-mesoscale domain, which may eventually form dense clusters of floating material. LWs typically take filamentous shapes a few metres wide and from tens to thousands of metres long[13,16], showing litter densities up to 10,000-fold higher than in their surroundings[17]. LWs can theoretically be used to flag episodes of severe surface pollution[13]. However, the use of these sparse and ephemeral targets for ML monitoring requires testing whether their abundance is sufficient to capture spatio-temporal patterns at any scale, and knowing the true meaning of this proxy.

The objective of the present study is twofold. First, it aims to define an optimal Earth-Observation (EO) mission concept for the global-scale monitoring of ML, referred to as the EO4ML mission. Second, the study conducts a proof of concept to test the feasibility and usefulness of EO4ML by implementing it with available technology on a real-world case study. On the basis of the optimal EO concept, we derived a detector for use with the multispectral acquisitions of the in-orbit Copernicus Sentinel-2 sensor. This suboptimal detector was applied on a continuous basis throughout the Mediterranean Sea, selected as our region of interest for its notable socio-ecological value and high ML pollution[18].

## Results

### Optimal concept and proof of concept

Plastic materials have an unequivocal human origin and are predominant among floating ML[6]. Therefore, the definition of the optimal mission concept for surface ML monitoring used plastic debris as the core target (Fig. 1). In the first step, reflectance spectra were obtained for varying concentrations and plastic polymers, then incorporating then other materials relevant to the radiative signal of floating litter[19,20] (Supplementary Information [SI] Sections S1.2.1 and S3.1.1, Fig. S3). This spectral library was fed into an atmospheric radiative transfer model in order to determine spectral radiances at the top of the atmosphere. The isolation of spectral bands for the sensor and their width was achieved by optimising the remote sensing performance (as signal-to-noise ratio) with changing fractions of plastic polymers over a range of observation conditions (SI Sections S1.2.2 and S3.1.2, Figs. S13 to S21). A set of additional bands was configured to specifically address the quantification of other abundant floating materials (i.e. algae, driftwood and seafoam) and to support atmospheric correction and cloud detection (Table S3). This exercise allowed us to define functionalities for a total of 23 candidate spectral bands (Table 1, Figs. S22 to S24).

The minimum threshold requirements for an EO4ML mission would involve a Super-Spectral Instrument with at least four bands in the near-infrared and two bands in the short-wavelength infrared, a configuration resulting from the significance of infrared radiation in the spectral signature of plastic. The proposed sensor could provide true detection from the natural sunlight reflectance of an ocean plot with a minimum surface plastic coverage of 1% (SI Section S3.2.1). However, this threshold is unlikely to be achieved extensively over the ocean surface[11,12], so even an optimal sensor would be obliged to focus on detecting LWs, and use a pixel size at least in the typical range of LW widths (~10 m) to increase the probability of exceeding the 1% threshold. By setting the spatial resolution at 10 m, a two-satellite constellation would achieve global coverage with a revisit time of a few days, a time resolution benchmark for many of the applications identified by the community[10].

The in-orbit Multi-Spectral Instrument of the EU Copernicus Sentinel-2 mission (S2-MSI) presents some compatibility with the technical requirements of an optimal EO4ML mission concept, in particular spectral bands within the infrared region with medium-high spatial resolution (10–20 m). These features provided us with the opportunity to carry out a proof of concept (PoC) to test the applicability of LWs as a proxy for ML monitoring (Fig. 2). By adapting an

**Table 1 | Optimal threshold specifications for the EO4ML mission**

| Instrument | Super-Spectrometer (SSI) |
|---|---|
| Field of View | 20.6 degrees (ca. 290 km projected on the surface according to orbit) |
| Maximum view zenith angle | 15 degrees |
| Orbit | Sun-synchronous at 786-km altitude (ca. 100 min period) |
| Coverage | Global |
| Geometric revisit time | 5 days (with a two-satellite constellation) |
| Spectral bands | 23 in total. 8 in VIS, 11 in NIR, and 4 in SWIR |
| Spectral range | 0.4 µm –2.4 µm |
| Optimal bands | |
| for detection | 826 nm, 913 nm, 1100 nm, 1515 nm, 1782 nm, 2031 nm |
| for false positive discrimination | 510 nm, 972 nm, 1205 nm, 1325 nm |
| for clouds and atmosphere | 400 nm, 780 nm, 945 nm, 1375 nm |
| Required signal-to-noise ratio | Minimum average of 300 for VIS, 200 for NIR, 100 for SWIR |
| Minimum spatial resolution | 10 m |
| Minimum plastic content | 1% of surface cover within the pixel |

To improve the signal-to-noise ratio, orbital altitude would be potentially reducible and/or opting by increasing integration time. Lowering the orbit will improve spatial resolution at the expense of reducing effective swath and having a longer revisiting time. Thirteen spectral bands are highlighted here. Functionalities for 23 bands useful in an EO4ML mission, including those related to the spectral unmixing of LW components, are described in Table S23. VIS, visible; NIR, near-infrared, SWIR, short-wavelength infrared.

infrared spectral index for this sub-optimal sensor (SI Section 2.1.7), the minimum plastic surface coverage required to achieve detection was estimated to be close to 20% (SI Section 2.4, Fig. S9). However, other drifting materials commonly included in the LWs such as weeds, driftwood or metastable seafoam, may also contribute to the S2-MSI detections[21]. Thus, the most reasonable option for the detector in the PoC was a probabilistic dichotomous (presence/absence) classification of image pixels with or without plastic-like spectral profiles. In addition, the detector applied a strict contextual filtering (SI Section 2.1.8) to deal with the spectral noise in the S2-MSI measurements[21]. Thus, only those clusters of pixels with positive detection, resembling filament-shaped patches longer than 70 m, were taken into account.

While spectral detection was targeted on plastic, PoC detections must be considered here as ML aggregations containing varying fractions of both anthropogenic and natural debris due to the known limitations of the S2-MSI[21]. Eventually, with dedicated spectral bands (Table 1), it will be possible to disentangle the contributions of the main litter components (e.g. plastic, weeds, driftwood, metastable seafoam) and estimate plastic concentrations per pixel. Importantly, an optimal sensor would also allow a proper account of small scattered litter patches (<70 m), which should result in a significant increase in the number of detections and the capability for defining spatio-temporal patterns.

## Marine litter windrows

A total of 14,374 LWs were detected and supervised in the 75-month monitoring of the Mediterranean Sea from July 2015 to September 2021 (Fig. 3). To date, only a few hundred LWs have been characterised, mainly because of the difficulties encountering these short-lived structures at sea[13]. The 14,374 LWs covered 94.5 km² of sea surface, an area equivalent to 7,500 football fields. The size distribution of the filaments detected from space agreed with reported measurements from ships[13,16], predominantly about 1 km long (Fig. S28). Nevertheless, remote detections also included unprecedented LW lengths. Twenty-

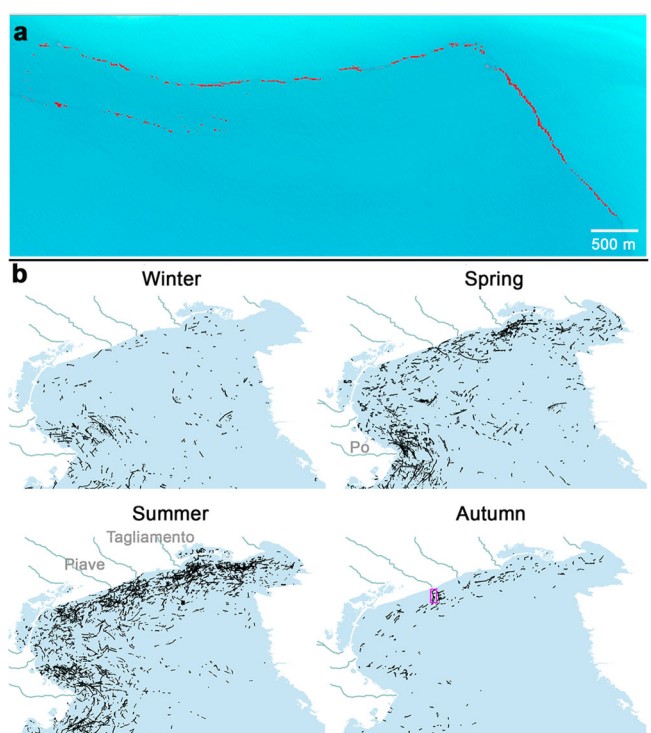

**Fig. 2 | Detections of litter windrows (LWs) in the North Adriatic Sea. a** LW seen on a Sentinel-2 L1c true colour image (31 October 2018, off Piave River, Italy). Pixels with plastic-like spectral profiles are coloured in red. These pixels usually appear in a patchy pattern along the LWs. The complete set of images automatically generated by the detector for each single filament is shown in Fig. S25. **b** All filaments detected per season in the innermost section of the North Adriatic Sea. The main rivers are outlined, and a pink box marks the LW in the image above. The monthly series of LW maps for the entire Adriatic Sea is provided in Fig. S26.

seven filaments (0.2% of the total) were longer than 10 km, up to 23 km long.

There is a wide array of hydrodynamic processes able to generate the submesoscale convergence structures necessary to form LWs[13]. Several authors pointed to surface fronts associated with internal waves as the predominant mechanism of LW formation[22,23]. The orientation of LWs in the Mediterranean, mainly parallel to shoreline and isobaths (Fig. 2b), fits with this proposal[23]. Notwithstanding, other convergence processes like estuarine fronts, open-ocean density fronts and Langmuir circulation were also apparent among our detections.

## Litter windrow density

The raw output of the detector used in the present PoC is a categorical classification of pixels, with either the presence or absence of dense litter cover, mapping LWs at a spatial resolution of 10 m. Quantitative assessment of ML pollution on a continuous scale was done through the fraction of sea surface area covered by LWs, referred to as litter-windrow density (LWD, in ppm; e.g. m² per km²). Importantly, LWD was calculated solely over the sea surface under environmental conditions deemed suitable for litter aggregation, i.e. for winds lower than 5 m s⁻¹ (see "Methods"). By reducing spatio-temporal bias due to different wind conditions and, consequently, varying conditions for LW formation, this wind-based constraint was intended to improve the comparability of LW detections in space and time, an issue hitherto unaccounted for.

On average, LWD for the whole Mediterranean was 0.2 ppm. For reference, the fraction of the Mediterranean Sea surface covered by plastic debris measured with surface-trawling nets is about 1.0 ppm

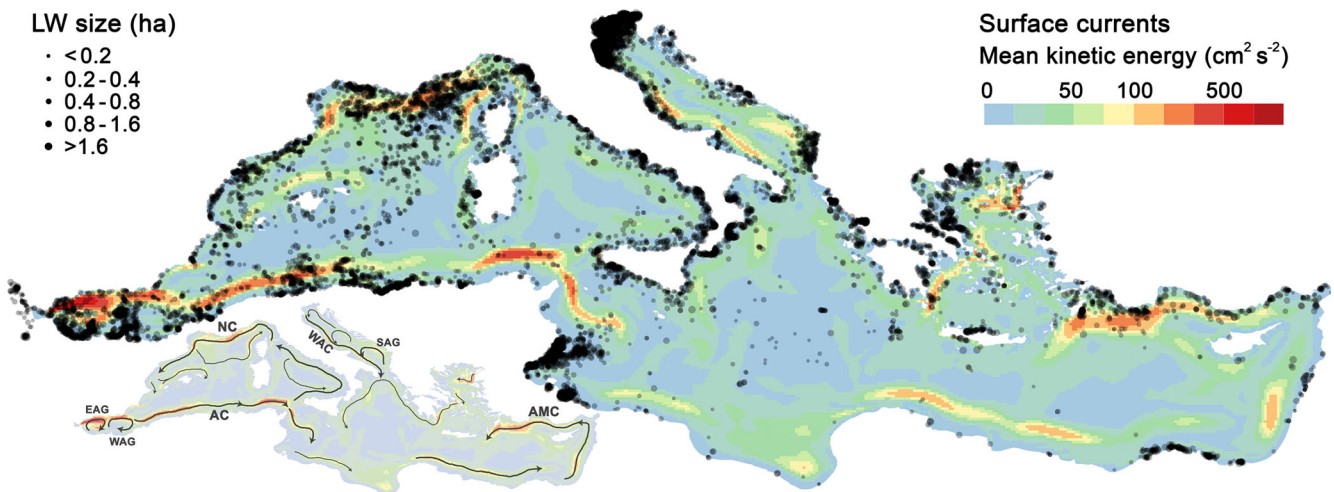

**Fig. 3 | Litter-windrow detections in the Mediterranean Sea.** The size of the semitransparent black dots is proportional to the LW area. Background colours show the mean kinetic energy (MKE) resulting from sea surface circulation patterns[55]. MKE increases where surface currents are stronger. The inset below indicates the main currents and gyres on the MKE map: Eastern Alboran Gyre (EAG), Western Alboran Gyre (WAG), Algerian Current (AC), Northern Current (NC), Western Adriatic Current (WAC), Southern Adriatic Gyre (SAG), Asia Minor Current (AMC). Selected areas are zoomed in Fig. S29.

(i.e. $m^2$ of plastic per $km^2$ of sea surface, accounting for both scattered and clustered plastics)[12]. LWD decreased exponentially with distance to shore, suggesting a strong control of the release of litter from land on the distribution of LWs (Fig. 4). Nevertheless, this decreasing trend showed consistent irregularities, mainly a litter build-up in the 5 to 30 km distance to land, which persisted seasonally (Fig. S30). Furthermore, the LWD peak at around 10 km matched the maximal LW lengths. This litter build-up seems to be the result of a bottleneck in the seaward propagation of litter, most likely caused by the strong currents commonly flanking the shoreline, the so-called boundary currents. In the Mediterranean Sea, they form a counterclockwise circulation belt that reaches, on average, its maximum strength at 50 km from land (yellow line in Fig. 4). Field samplings have already reported the blocking effect on the offshore transport of litter in the Northern Current, the regional boundary current in the north-western Mediterranean[24]. In our satellite images, this effect was particularly evident when coastal jets cut through highly littered waters (e.g. off the coast of France), or in the low permeability of the steady gyres placed in sub-basins such as the Alboran and Adriatic Seas (Figs. 3 and 5).

## Control of spatial patterns

The basin-scale mapping of LWD showed hotspots scattered predominantly in nearshore waters, particularly in the western and central Mediterranean (Fig. 5). The most intense hotspots were located in the south-western Alboran Sea ($LWD_{mga}$ = 9.1 ppm), Algerian waters (12.9 ppm), Gulf of Gabes (15.8 ppm), off Calabria (south-west Italy, 9.0 ppm), and, most notably, at the northern end of the Adriatic Sea (55.8 ppm; a region-specific analysis is presented in SI Section S4). In contrast, the south-eastern basin had the most extensive clean waters, which is consistent with the basin-scale surveys conducted for both microplastics[12] and human-made mega-litter[25].

Most models of land-to-ocean plastic inputs (PI) predict an opposite spatial distribution of sources if compared to our LWD map or to previously reported patterns of floating plastic[12,25], which increase from the south-eastern to the north-western Mediterranean. Current PI models rely on estimates of mismanaged plastic waste (MPW) on land, assuming that the efficiency of MPW transfer from land to ocean ($\eta$) is invariant in space[1] and time[26]. Therefore, these models predict that the north-western shores, with high economic status and presumably effective waste management, emit little plastic[1,2]. However, field observations provide growing evidence of a strong control of environmental drivers on litter emissions, namely precipitation and surface runoff[27–29]. Introducing environmental control over $\eta$, PI predictions were significantly correlated with nearshore LWD (Fig. 6a, Fig. S31). Lack of precipitation in the south-eastern watersheds (<100 mm $y^{-1}$) accounts for the relatively low LWD in their adjacent waters. On the other extreme, watersheds on the north-western shores receive up to 30 times more precipitation, especially on the Alpine watersheds in Italy, connected to the major LWD hotspot in the Mediterranean (Fig. 5).

The spatial correlation found between PI and LWD implies that much of the litter remains near its land-based source. In this regard, boundary currents were presented in the previous section as important transport barriers. However, these barriers are also susceptible to both along-shore and cross-shore transport, leading to important deviations in the PI-LWD correlation (Fig. 6). First, the counterclockwise circulation of the boundary current system may entail relevant advection alongside the coastline. Major alongshore pathways appeared to run from the North Adriatic hotspot through a corridor along the Italian eastern coastline, from north Tunisia to the Gulf of Gabes, and through the Asia Minor Current to the Rhodes Basin (Figs. 3 and 6b). Secondly, floating litter may eventually cross-boundary currents, especially in events of massive PI and strong seaward winds[24]. ML release into open waters was particularly evident in the Sea of Sardinia, receiving inputs from France and Algeria (Fig. 5). The most striking seaward surge of LWs occurred in spring 2018 (Fig. 7b), coincident with record-breaking rainstorms and flooding along the northwestern Mediterranean coast (southern France and northeastern Spain)[28,30]. The meteorological conditions during this event, as well as field surveys during similar events[24], point to the northwesterly wind flow (Mistral) as the driver of this massive transport of litter, intruding more than 300 km offshore. Mistral is the strongest wind system in the Mediterranean[31], and it is possibly involved in the above-normal LWD observed in the western waters of Corsica and Sardinia and off the French Riviera (Fig. 6b). Anyhow, despite the occasional relevance of along-shore and cross-shore transport, the PI-LWD linkage remained broadly significant at basin scale. With due caveats, this spatial correlation suggests a primary relationship of LWs with nearby (<150 km) litter sources and, particularly, with plastic emissions (Fig. S31). Plastic litter is highly persistent, in fact, the majority of all floating items in the Mediterranean Sea are

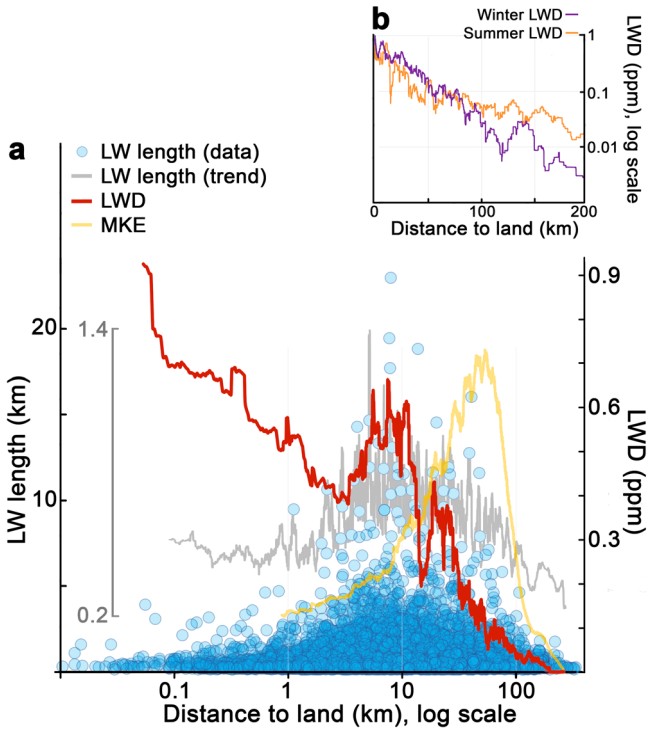

**Fig. 4 | Litter-windrow length and litter-windrow density (LWD) in relation to distance to land. a** The blue dots indicate the length of each LW (left axis), while the grey line shows 100-data moving averages for the dot cloud (inner grey scale). The red line corresponds to moving averages for LWD on subsets totalling $10^5$ km$^2$ of sea surface (right axis). The same filter was applied to sea-surface mean kinetic energy[55] (MKE, yellow line), ranging from 3 to 45 cm$^2$ s$^{-2}$ (scale not shown). **b** Seasonal patterns of LWD in relation to distance to land. Extended summer covers spring and summer. Extended winter comprises autumn and winter, covering the high wind season[40]. While panel (**a**) addresses LWD irregularities in nearshore waters by graphing distance to land in log scale, (**b**) highlights the basin-scale distribution of LWD from nearshore to offshore areas. The seasonal patterns of LWD in relation to distance to land on a log scale are shown in Fig. S30.

plastics (> 75%, accounting for both anthropogenic and natural items)[32,33].

## Control of temporal patterns

The dynamics of ML hotspots were characterised by pronounced pulses, often associated with record rains and floods over land (Fig. 7). Short and strong LWD peaks every 2 to 3 years generally accounted for more than half of all LW detections, suggesting that LW formation is particularly reactive to extreme rainstorm events.

Rainstorm systems typically show a diameter of a few hundred kilometres and travel thousands of kilometres across the Mediterranean during their lifetime (2-8 days)[34]. On a basin-wide scale, the pulsating LWD dynamics induced by these cyclones were dampened by the multiplicity and asynchrony of local pulses, but basin-wide LWD also trailed precipitation and runoff trends (Fig. 8a). The rainiest periods were followed by increases in LWD, mainly in autumn 2016, spring and autumn 2018, and spring 2021. The main uncoupling between precipitation and LWD was found in the second half of 2019 (high precipitation and decreasing LWD). In the previous year, both precipitation and LWD reached the highest values in the time series, twice as much as in other years, which agrees with the annual litter inputs measured in rivers[28,29]. The intense flushing of watersheds during the wet 2018 could have depleted the MPW available for runoff transport and thus explain the weak response to precipitation in 2019. When rivers overflowed in 2018, field surveys reported how MPW was

massively conveyed to the sea, to inland trapping sites from where it is difficult to re-mobilize, or even buried[27–29,35].

At a seasonal scale, rainfall in the Mediterranean typically extends from early autumn to spring[36]. In contrast, the seasonal pattern of LWD was bimodal, with peaks in autumn and spring separated by a steep winter minimum (Fig. 8b). The relative importance of the two seasonal LWD peaks changed from west to east, in accordance with the timing of the cyclonic systems hitting each sub-basin. Spring rainstorms impacted mainly the western Mediterranean[30], triggering the highest LWD peaks. The Atlantic Ocean supplies many of the cyclones entering the Mediterranean, resulting in a longer and more intense rainy season in the west[36]. Likewise, cyclogenic activity within the Mediterranean takes place mainly over the western and central sub-basins[34]. In our observation period, the autumn formation of storms and tropical-like cyclones (e.g. Zorbas medicane[37]) controlled the seasonal LWD pattern in the central Mediterranean. In the south-eastern Mediterranean, the lack of rainfall and reduced runoff were apparently the main reasons behind the low LWD, with only small coastal hotspots (e.g. Nile delta, off Lebanon). The weak connection to land-based sources was already mentioned to explain the extraordinary paucity of microplastics found in the south-eastern Mediterranean[12].

Unlike the spring and autumn peaks, the winter decline in LWD was highly consistent year to year over the entire basin, being captured by ship-based litter monitoring[38,39]. Seasonal wind speed maximum is also highly steady throughout the entire Mediterranean Sea[40], matching the LWD minimum from December to January (grey dotted line in Fig. 8b). Given that LWD is calculated only over the sea surface suitable for LW formation (winds <5 m s$^{-1}$), wind effect should mainly be reflected in LWD through the advection of floating litter or its removal from the sea surface by beaching. In the North Adriatic, floating litter was systematically cleared every year during the wind peaks in December and January (Figs. S26 and S32). We hypothesised an increased wind-driven dragging of surface litter towards coastal waters and onto the shores in winter. This onshore transport of floating litter was supported by a clear seasonal shift in the spatial distribution of LWs over the Mediterranean. In summer, LWD presented a smooth profile with distance to land, while offshore pollution decreased and litter retreated to coastal waters in winter (Fig. 4b). LWD decreases in the Mediterranean Sea during the high wind season, but winter LWD exceeds the summer LWD in the water strip closest to shore (<50 km from land). Seasonal patterns of beach litter are difficult to unravel because they are strongly influenced by the waste input from summer beach tourism. However, considering only data from non-tourist beaches in the Mediterranean, beach litter loads corroborated the hypothesised increase in litter stranding towards winter[41,42].

## A new scenario for research and management

The PoC demonstrated the potential of an EO mission using sub-mesoscale surface aggregation structures as a proxy for monitoring ML pollution. Despite the sensor limitations used in the PoC and the ephemeral nature of the LWs, the number of detections gathered over a 6-year period proved sufficient to delimit mesoscale accumulation zones and capture interannual and seasonal patterns. Maps and trends of LWD showed high consistency with available field data[12,25,29,38,39,41,42], as well as with both environmental (P, wind, MKE) and anthropogenic (population, MPW) drivers, composing an unprecedented basin-scale picture of the source-to-sink ML dynamics. Land-based mismanaged waste and torrential rains controlled the inputs of ML, while coastal boundary currents and wind-driven surface sweep were the key drivers for its distribution over the sea surface.

LW detections were primarily related to the location and magnitude of land-based ML inputs during the previous weeks. This feature renders the LW proxy particularly helpful for surveillance and management purposes. Using real case scenarios, Supplementary

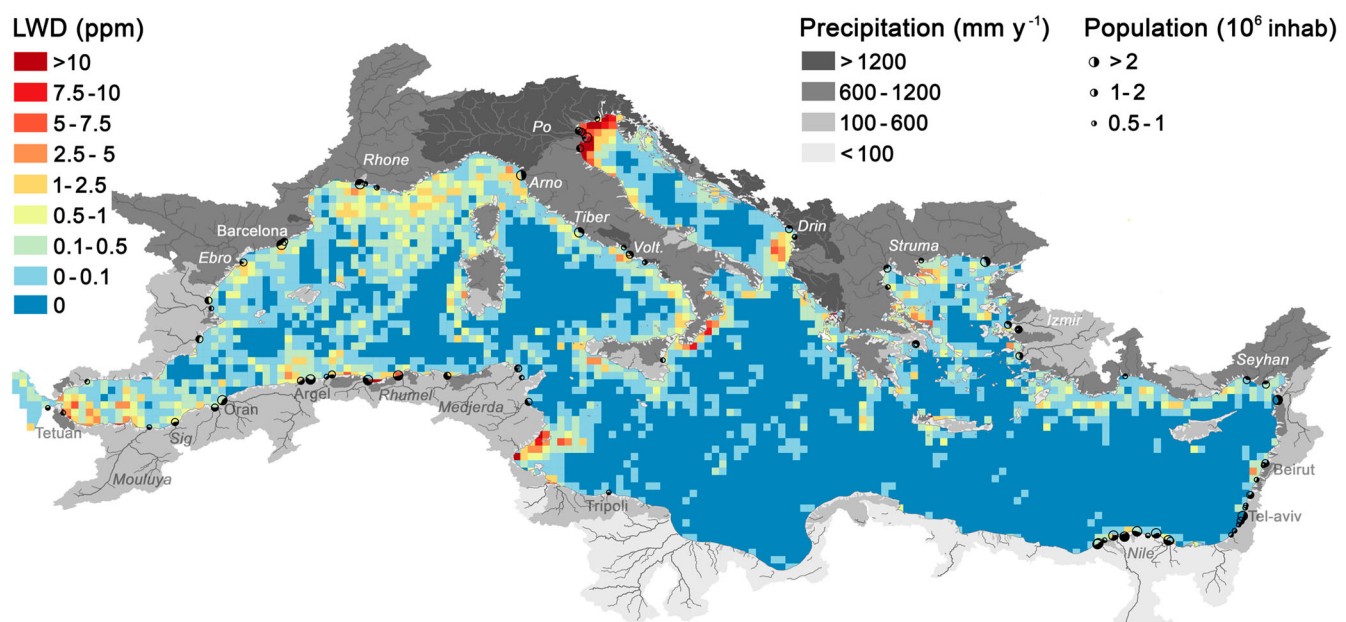

**Fig. 5 | Time-averaged litter-window density (LWD) in the Mediterranean Sea.** The most populated watersheds are marked at their sea outlet, while the mean annual precipitation per watershed is shaded in greyscale (see legend). Relevant population centres and rivers are labelled.

Information illustrates some applications, such as the identification of pollution hotspots, shipping-related sources of litter, long-term evaluation of action plans or better targeting of clean-up efforts (SI Section 4, Fig. S33). The readiness level of the present monitoring tool allows for its implementation in other ocean regions; however, there is still much room for improvement. S2-MSI surveys in other ocean regions could advance the mission concept, analysis methods and data interpretation. Current LWD estimates account for wind conditions since LW formation requires not only high ML load but also favourable sea state[13]. Nevertheless, further refinement of LWD estimates should ideally account, along with disruptive forces (*e.g.* wind mixing), for major formation forces (e.g. tidal forcing).

The next steps should also advance on the ground truthing (Fig. 1). The current matching of satellite detections and field observations is limited to fake targets (artificial LWs)[15,20] and reports of dense LW sightings[14] (Figs. S9 and S11). Filling this gap requires targeted and coordinated research on the LW phenomenon and the combination of several approaches. Fake targets are an effective approach, which might be further extended to different LW compositions and environmental conditions. Matching detections of natural LWs and field measurements is a more challenging task due to the difficulty of finding LWs from ship surveys[13]. We can now take advantage of satellites to address areas and periods of common LW formation (Fig. 5). Complementary observation platforms with larger coverage areas than ships (e.g. long-range drones, aircrafts) could also significantly increase LW findings[25]. Additional inputs for calibration and validation can be gained from the ocean-wide spreading of animal-borne cameras[43], particularly from species using windrows as feeding spots. Citizen science could also contribute with opportunistic observations whereas from leisure vessels, fishing, or commercial ships. Finally, another pivotal approach stems from the overlapping of S2-MSI detections with observations from medium-to-high-resolution hyperspectral satellites[44].

The sustained monitoring of LWs from space can be a game changer for ML research and management and could also be of interest beyond the ML problem. LWs trace currents, retention zones and repelling structures from submeso- to macro-scale, opening new prospects for understanding the multiscale interplay of ocean transport. A relevant ecological implication of LWs relates to their role as hubs of marine life on the ocean surface, generating aggregations of small organisms and attracting large predators (e.g. fish, seabirds)[13,23]. The main limitation of the in-orbit S2-MSI for ML monitoring is its sensitivity, since it requires long and dense LW structures to provide reliable detections. An optimal sensor could reduce the detection threshold by one order of magnitude, as well as supporting the unmixing of LW components. Accounting for litter components would diversify the application possibilities beyond plastic-specific monitoring (e.g., metastable seafoam in eutrophication). Side applications derived from targeting metre-sized items on the ocean surface might cover areas such as cargo loss, oil spills, navigational safety, or search and rescue at sea. Overall, the concept of an EO mission dedicated to ML monitoring proves to be not only plausible but also highly promising.

## Methods
### Optimal mission concept
The aim of the mission conceptualization was to establish the minimum base requirements for an EO mission founded on a spectral analyser instrument with dedicated bands for marine plastic litter (EO4ML mission). To do so, we carried out a combined analysis of user needs, monitoring scenarios as well as ML physical properties potentially useful for space-based sensors. Both observational capabilities and technical requirements were contrasted with existing and near-future technologies. The data for this exercise were generated in laboratory, field, and through modelling tests. Ultimately, the trade-off analysis provided the key theoretical specifications for the EO4ML mission, particularly on the identification of candidate spectral bands and signal-to-noise ratios. The full process for defining the optimal mission concept is described, step by step, in Supplementary Sections S1 and S3.

### Proof of concept
The PoC was based on the finding of limited compatibility between the spectral bands identified for an optimal mission and the infrared bands of the Multi-Spectral Instrument of the Sentinel-2 mission (S2-MSI). The EU Copernicus Sentinel-2 mission takes images over a grid with $100 \times 100$ km$^2$ tiles and a spatial resolution up to $10 \times 10$ m$^2$. The revisiting time for the Mediterranean is 2-3 days, with two satellites operating simultaneously (since 7 March 2017). For the present PoC, a

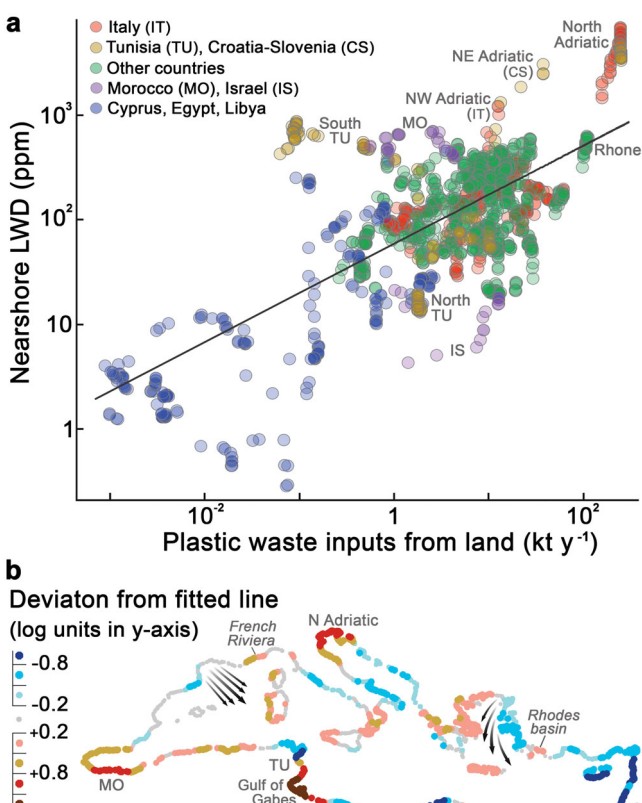

**Fig. 6 | Relationship between modelled plastic inputs from land (PI) and litter-windrow density (LWD) in adjacent waters. a** PI-LWD scatter plot in which each dot corresponds to a single watershed draining into the Mediterranean (*n* = 1,562). Circular areas (150 km radius) centred on the outlet of each watershed were used to calculate LWD in adjacent waters. **b** Location map of watershed outlets, with colours indicating the sign and magnitude of the deviation from the fitted line in the PI-LWD plot (r = 0.703; p < 0.00). Deviations can be partly related to surface transport by wind and water currents. Watersheds above the line (reddish dots; e.g. Gulf of Gabes in south Tunisia) are associated with adjacent waters with more litter than expected from PI, being potentially related to coastal accumulation or import; and conversely for watersheds under the line (bluish dots; e.g. north Tunisia). Arrows in the map indicate the most persistent wind systems in the Mediterranean Sea, Mistral (Gulf of Lyon) and Etesian winds (Aegean Sea)[31].

total of 288,166 images (150 TB of data) were processed in a High-Performance Computing facility, corresponding to 411 tiles covering the entire Mediterranean Sea and for the period from 4 July 2015 to 21 September 2021 (75 months).

Building on the knowledge gained in the optimal mission concept, we developed a specific spectral index to classify S2-MSI pixels with and without plastic-like signatures (SI Section 2.1.7). After the spectral classification of individual pixels, the detector performs filtering to select only those aggregations of pixels with positive detections resembling filament-shaped patches over 70 m long, with the aim of minimising the number of false positives (SI Section 2.1.8). In a later step, the image clippings of each candidate filament automatically detected by the detector were supervised by a team of 6 researchers (SI Section 2.3.3). Six different contributors to false positives were recognized in the supervision, i.e. aeroplane exhausts, cloud edges, wave glint, ship trails, human-made structures and image stitching (Figs. S34 to S39). LW clippings are archived in an open repository for future development of automatic classification methods (https://doi.org/10.5281/zenodo.11045944).

Supplementary Section S2 provides a detailed description of the PoC detector (image pre-processing, spectral indices, and filament

contextual identification), detector tests, and supervision of the results.

## Litter windrow density

The raw output of the PoC detector (i.e. filaments with dense ML coverage) was expressed as a quantitative variable using the area of LWs relative to the observed sea surface suitable for litter aggregation. Mathematically, this litter-windrow density (LWD, in ppm) was calculated as:

$$LWD = \left(a_{lw}/a_o\right) \cdot 10^6 \qquad (1)$$

where $a_{lw}$ is the surface area of LWs, and $a_O$ is the total sea surface area observed by the sensor (not covered by clouds) and deemed as suitable for LW formation. In this regard, we removed from $a_O$ the sea surface area associated with wind speeds higher than 5 m·s⁻¹. Neutral wind speeds at 10 m above the sea surface ($u_{10N}$) were acquired from the ERA5 database[40]. We found LWs with $u_{10N}$ up to 11.2 m·s⁻¹, however, the probability of detection dropped sharply above 5 m·s⁻¹ (Fig. S27). More than 95% of all detections were below this threshold, likely due to vertical and horizontal mixing by wind and wave action. Interestingly, the threshold of 5 m·s⁻¹ was also defined as that above which vertical mixing of floating plastic debris becomes significant[45].

For our study scale, we selected a 0.25-degree latitude-longitude grid as the standard binning for LWD computation. Note that the average LWD for a large region (e.g. mesoscale ML hotspot) is independent of the grid selected, but finer grids (smaller cells) result in larger statistical dispersion of LWD values across grid cells, with lower minima and higher maxima.

Apart from the average statistic, we used the maximal gridded average (LWD_mga) to characterise mesoscale ML hotspots. LWD_mga corresponds to the spatial maximum of the annual averages in the 0.25-degree latitude-longitude grid over the referenced region. The annual average for each grid cell was estimated by averaging the means for the 12 months of the year, using data over the entire study period. Cells containing a sea area of less than 100 km² were discarded for LWD_mga, as excessively small sea areas may be unrepresentative and reach extreme maxima (> 1,000 ppm). Unlike the spatial average, LWD_mga shows the advantage of being less dependent on where the boundaries of the hotspots are placed.

## Spatio-temporal drivers

Given the paucity of field data with high spatio-temporal resolution, the consistency of the LWD observations was in large part tested on its relationships with environmental drivers and the production of mismanaged plastic waste (MPW). This was done separately for the spatial and temporal variability, using the time-averaged map and the basin-wide time series, respectively. The time-averaged map of LWD has the advantage of a weak dependence on the temporal response to drivers such as rainfall, which could include time lags or gaps in response to the eventual depletion of land-based MPW. In the case of the LWD time series for the entire Mediterranean basin, we took advantage of the fact that they are independent of hydrodynamics and surface transport of litter.

## Spatial analyses

The time-averaged map of LWD in the Mediterranean Sea was contrasted with available models of plastic emissions from land. For this purpose, the first step was the exploration of the performance of the main existing models of land-based production of MPW (MPWP), as well as possible upgrades (Table S32). MPWP (in t y⁻¹) for watershed *i* is

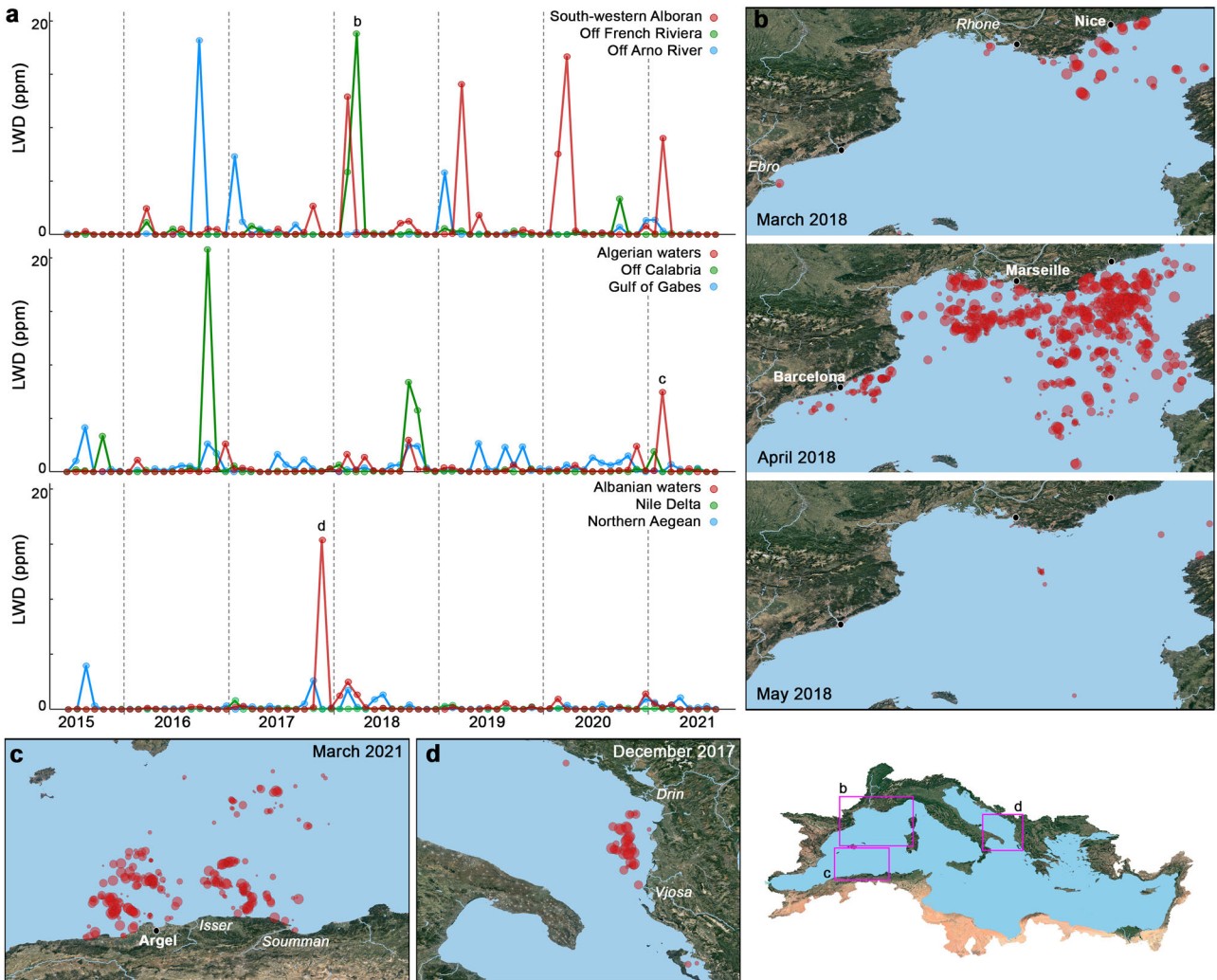

**Fig. 7 | Monthly litter-windrow density (LWD) in marine litter hotspots.**
**a** Monthly LWD series in nine mesoscale hotspots. The main pulse in the French Riviera (top chart), Algerian waters (middle chart) and Albanian waters (bottom chart) are marked with the letters **b**, **c** and **d** and illustrated on the surrounding maps. **b** Monthly distribution of LWs in the north-western Mediterranean in March, April and May 2018, covering the highest LWD pulse in waters off the French Riviera. LW detections are represented by semi-transparent red dots of size proportional to the LW area. **c** LW distribution during the main pulse in Algerian waters in March 2021. **d** LW distribution during the main pulse in Albanian waters in December 2017. Mapped areas are marked with pink rectangles on the lower right map. The main LWD pulses were generally related to news of record rains and floods in the media, such as those in the north-western Mediterranean in April 2018[56], in Algerian waters in March 2021[57], and in Albanian waters in December 2017[58].

commonly estimated as:

$$\text{MPWP}_i = p_i \cdot g_c \cdot f_c \cdot (in_c + l) \qquad (2)$$

where $p_i$ is the population in watershed $i$ (inhab, restricted to the population living within 200 km from the coastline); $g_c$ is the per-capita waste generation rate for country $c$ where $i$ is located (t inhab⁻¹ y⁻¹); $f_c$ is the fraction of plastic in the waste stream in $c$; $l$ is a common (country-independent) fraction of littered waste; and $in_c$ is the fraction of inadequately managed waste for $c$, which is based on the economic status of $c$. A total of 1,562 watersheds were delimited within the Mediterranean basin from the HydroSHEDS database[46]. $p_i$ was estimated from high resolution (0.5 arc minutes, ~1 km) gridded population density data[47]; while $l$, $g_c$, $f_c$ and $in_c$ were obtained from the model by Jambeck et al.[1]. ($J_{WB}$), and from the low, mid and high-range models by Lebreton and Andrady[48] ($LA_L$, $LA_M$, $LA_H$, respectively). We also tested a version of Jambeck's model based on data from Waste Atlas[49] ($J_{WA}$ model) instead of that of the World Bank[50]. Moreover, an additional model was only scaled with the population size, following

the suggestion by Weiss and coworkers[51] (W model). For this last approach, a general per-capita MPW generation was applied (i.e. 13.5 kg inhab⁻¹ y⁻¹), corresponding to the global average of the mid-range model by Lebreton and Andrady[48]. Note that the election of this per-capita rate conditions the magnitude of MPWP but not its relative spatial variability. In all, six different models were explored for MPWP (Table S32).

Land-to-ocean transfer rate of MPW ($\eta$, in y⁻¹) is likely key for understanding marine plastic pollution, but little is known about its magnitude and spatio-temporal variability. Here, we used precipitation and surface runoff as drivers of $\eta$, as proposed by a growing body of field studies[27–29]. Precipitation and runoff at a spatial resolution of 6 arc minutes (~10 km) were obtained from the ERA5-land dataset[52]. Both variables (expressed in mm y⁻¹) were converted into $\eta$ units (y⁻¹) by scaling to a Mediterranean basin-wide average of 0.25 y⁻¹, corresponding to the global mid-range $\eta$ suggested by Jambeck and coworkers[1]. Again, this linear transformation was just intended to maintain the dimensional coherence, having no effect on its relative variability.

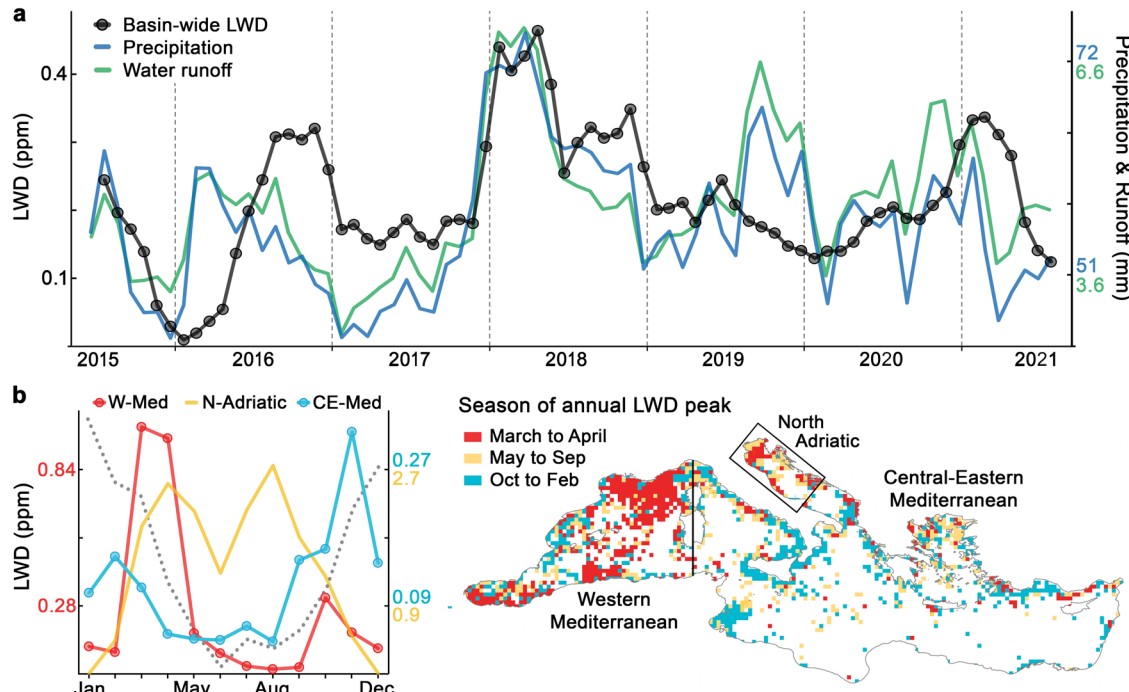

**Fig. 8 | Interannual and seasonal patterns of litter-windrow density (LWD) at basin scale. a** Interannual trend of LWD in the Mediterranean Sea, and trends of precipitation and surface runoff over the drainage basin. Linear correlation of LWD to precipitation and runoff was statistically significant ($p < 0.01$) for lags from 0 to 4 months, being maximum at 1-month lag for both precipitation ($r = 0.586$) and runoff ($r = 0.523$). The largest decoupling was in 2019 (see text). **b** Averaged seasonal patterns of LWD for major covarying regions, Western Mediterranean (W-Med), Central-Eastern Mediterranean (CE-Med) and North Adriatic (N-Adriatic). Note the different LWD scales (see colours). The Grey dotted line is the fraction of the Mediterranean Sea surface exposed to winds over $5\,m\,s^{-1}$, ranging from 63% (January) to 39% (June)[40]. The right map shows the season of annual LWD peak per 0.25-degree cells. The division between W-Med and CE-Med was made from the meridian along Corsica and Sardinia[34], while the N-Adriatic is framed into a rectangle. The monthly LWD series for each region is shown in Fig. S32.

Plastic input into the ocean (PI, t y$^{-1}$) in watershed $i$ was calculated as the product of MPW (t) and $\eta$ (y$^{-1}$):

$$PI_i = MPW_i \cdot \eta_i \qquad (3)$$

For simplicity, $MPW_i$ available for runoff transport was assumed to be that generated in the current year (i.e. $MPWP_i$, Eq. 2), in accordance with previous PI models[1,2].

The analysis of the correspondence between potential explanatory variables (MPWP, $\eta$ and PI) and the spatial LWD pattern was addressed on the nearshore waters in order to reduce the role of hydrodynamics and transport of floating litter. Since LWD off a watershed outlet also depends on the plastic emissions from neighbouring watersheds, a range of circular buffers of different sizes was used to calculate LWD and independent variables (MPWP, $\eta$, PI). Centred at each single watershed outlet, we used circular buffers ranging from 10 to 200 km radius. Thus, LWD associated with a watershed outlet was calculated over the sea surface within the buffer zone, and PI was the sum of all the emissions from the watershed outlets within the buffer.

Finally, we performed a Pearson product-moment correlation test between nearshore LWD and predictive variables. A logarithmic transformation ($log(x + 1)$) was applied due to the wide range and skewness of the data analysed. The raw correlation (without log transformation) delivered higher Pearson coefficients (up to $r = 0.9$), however, this was due to an over-fitting of the high values, which minimizes the residual sum of squares but disregards data in the low range.

Of the six MPWP models explored (Table S32), the one based on population alone (W model) presented the highest spatial correlation with LWD (Fig. S31a). High range model by Lebreton and Andrady[48] (LA$_H$) also showed high Pearson coefficients ($r$), as it set the highest

fraction of littered waste ($l = 10\%$) and consequently gave the greatest weight to population in the MPWP estimate. This result suggests that factors other than economic status (e.g. consumer habits, beach tourism[41]) could also be playing a significant role in MPWP. Interestingly, the drivers to estimate $\eta$ (i.e. precipitation and runoff) were better predictors of LWD than MPWP. Lastly, as expected, PI achieved the highest correlations with LWD, in particular PI models based on a highly population-dependent MPWP (W·P and W·R models). The Pearson coefficients for the W·R model increased with buffer size, reaching the highest $r$ at 150 km radius (Fig. S31b). In order to use a model with both high $r$ and high spatial resolution, we selected the W·R model with 150 km buffers for the LWD spatial analysis (Fig. 6).

## Temporal analyses

Exploring the correlation between LWD and PI estimates over time involves further complexity due to the limited knowledge of the time variability of MPW available for runoff transport[1]. Accumulation or even depletion of land-based MPW could happen in particularly dry or wet years, respectively; and these changes likely also occur at seasonal or shorter scales due to the wide variability of both precipitation and surface runoff. Thus, for example, the first rains after summer could have a strong response in LWD, while later rains could have no effect on litter emissions.

Based on the strong control of precipitation and surface runoff on the spatial variability of LWD (Fig. S31), we opted to explore the monthly variability of basin-wide LWD from direct correlation with precipitation and runoff (as major drivers of PI). To address the complexity of the LWD response at seasonal or shorter scales, we extracted interannual trends of precipitation, runoff and LWD by subtracting seasonal components[53] and applying 5-month moving average filters. LWD changes may be also decoupled from precipitation and runoff

due to delays in river discharges and LW formation or to the accumulation of floating litter after consecutive months of rainfall. Estimates of residence time of litter on the sea surface usually range from weeks to months[54]. Hence, the relationship between the basin-wide trends of LWD and predictive variables (precipitation and runoff) was tested over a range of lag times from 0 to 6 months.

## Data availability

This paper provides a source data file, including the information associated with each of the LW detections obtained for the Mediterranean Sea from 4 July 2015 to 21 September 2021, namely location, date, size and distance to land. A single netCDF-4 file with all the detected LWs has been deposited at https://doi.org/10.5281/zenodo.11045944. This information is also available under free registration at https://bec.icm.csic.es/bec-ftp-service-registration. The raw Copernicus Sentinel-2/MSI L1c dataset is available at https://dataspace.copernicus.eu/. The ERA5 and ERA5-Land reanalysis datasets used in this study are available from the Copernicus Climate Change Service (C3S) at https://cds.climate.copernicus.eu/. Additional material related to this paper can be found at www.marinelitterlab.eu. Source data are provided in this paper.

## Code availability

The codes for this study are available upon request from the corresponding authors.

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

## Acknowledgements

This research was funded by a series of three contracts with the European Space Agency (ESA) over the period 2017 to 2022. It comprises the Activities entitled Remote Sensing of Marine Litter (RESMALI, ESA contract no. 4000121315 within the General Studies Programme), granted to M.A., A.C., G.B., and R.D.; EO tracking of marine debris in Mediterranean Sea from public satellites (ESA contract no. 4000124861 within the EO science for Society program), granted to M.A., A.C., S.A, G.S., and J.D.; and Mapping Windrows as Proxies for Marine Litter Monitoring from Space (WASP, ESA contract no. 4000130627, within the Discovery Element of the ESA's Basic Activities), granted to M.A., A.C., S.A, G.S., J.D., and R.S. The proof of concept was also supported by the Global Litter Observatory (CTM2016-77106-R/ AEI/10.13039/501100011033/ European Union – NextGeneration EU/PRTR), granted to AC, J.V., C.M. and D.G. M.A. and A.T. received supplementary funding from the Severo Ochoa Excellence Programme at the ICM-CSIC (Spanish Ministry of Science and Innovation), and from Ocean+ (European Union – NextGeneration EU) as part of the MITECO program for the Spanish Recovery, Transformation and Resilience Plan (Recovery and Resilience Facility of the European Union established by the Regulation (EU) 2020/2094). S.A. and G.S. received additional funds from the following projects: PRIN - EMME (contract 2017WERYZP), EURO-qCHARM (grant agreement 101003805), NAUTILOS (grant agreement 101000825) and SCOR FLOTSAM. The open access of this paper was co-funded by the QUALIFICA Project (QUAL21-0019, Junta de Andalucía) and the University of Cádiz.

## Author contributions

A.C. and M.A. conceived the study and wrote the manuscript. A.C., M.A., G.S., J.V. S.A., A.K., J.D., G.B., D.M., R.D., C.M. and D.G. carried out the experiments, modelling and field tests. A.C., M.A., J.D., G.B., R.D., R.S. and P.C. contributed to the design of the mission concept. R.S. designed and tested the contextual multiscale classifier of the detector. A.C., M.A., G.S., J.V. S.A., A.K., J.D., D.M., R.S., C.M. and D.G. contributed to the acquisition and analysis of data in the proof of concept. A.C., M.A., G.S., J.V. S.A., A.K., J.D., G.B., D.M., R.D., R.S., C.M., A.T., D.G. and P.C. contributed to the interpretation of results and reviewed the manuscript.

## Competing interests

The authors declare no competing interests.
