## [Peer Review File · Nature Communications]

Proof of concept for a new sensor to monitor marine litter from spaceREVIEWER COMMENTS

Reviewer #1 (Remarks to the Author):

Comments to ms by Cozar et al. (see also the annotated manuscript):

General evaluation: This manuscript presents interesting and potentially valuable results based on a new approach of detecting windrows from space, and confirming via spectral profiling that these windrows contain floating anthropogenic litter, in particular plastics. The general findings appear reasonable, namely more windrows in coastal regions, more litter windrows (LW) close to coastal sources and more abundant after rainfalls. Also the comparison of LWs with Plastic Inputs (PIs) appears suitable and generates very useful results (it basically is the Proof of Concept). I congratulate the authors for the idea and also for the immense work with the image processing and the data analysis.

While the key results (figures 1-7) are impressive and relatively easy to follow, the structure of the manuscript is confusing and especially the concepts presented in the text require revision and focusing. The authors present a number of constructs (e.g. the “PoC detection processor”) that are confusing in their logical conception and presentation. For example, they introduce the “Proof of Concept”, which certainly is a very useful and valid approach (and the PoC itself is not the problem). In the present context, the PoC could mean that they propose a stepwise approach that passes through the remote detection of windrows, then the spectral profiling of plastic litter (which identifies among all the detected windrows those that indeed contain large quantities of plastic litter and results in the LWs), and via quantification of size (length and area) of these LW, they then determine the LWD (in ppm, which seems to be pretty much “m² per km²”). In a next step they then correlate these LWDs with estimated Plastic litter Inputs (PI), which is based on input source (=rivers?), runoff, and population density. They find a reasonably good correlation between LWD and PI, which, of course, is not perfect, but they identify the areas where important deviations from the overall correlation are found, and they provide reasonable arguments to explain these deviations (more local processes that cause these deviations). All this together (correlation between LWD & PI; with explanation of major deviations) then could be considered as a first “Proof of Concept” for this new approach. This seems to be the underlying “concept” (sic) of this study, and the main results follow this line of argumentation and testing of the PoC.

However, the authors then create substantial confusion by introducing the “PoC detection processor”, which is a very odd construct. The PoC is here apparently converted to a specific “detector” that uses different spectral approaches and computing analysis steps to identify individual LWs. This (detection of Ws and identification of LW) is certainly an important part of this approach, but the linguistic construct of a “PoC detection processor” is very confusing and I strongly recommend to find a different construct for this, because the PoC is the umbrella concept that embeds the entire study: if the authors can show that their approach provides decent results that are useful to identify areas with high amounts of PL (accumulated in LW), then this is a reasonable Proof of Concept for their approach. In your analytical approach you certainly can have a detection processor, but I strongly recommend to give this a different

identify than “PoC detection processor”.

Another issue is the fact that the current approach only identifies LWs even though it seems that the authors might also be able to identify Windrows (Ws). I have to admit that I did not read the entire 137 pages of the electronic supplement, but I glanced through it and looked at different sections in order to better understand the overall approach. On P. 29 of the supplement, the authors say “To ensure our focus on LWs, we combine the spectral index with a contextual filter based on the geometry of such structures (section 2.1.8), and a further filtering was applied during post-processing (sections 2.2.2 and 2.3.3) to ensure that only LWs with high ML density were preserved for the PoC analysis.” In the context of this study, it seems very important to understand where windrows are and which of these windrows really contain ML, and maybe even how much ML. As discussed in some of my comments in the annotated manuscript, I suspect that the rainfalls flush a lot of natural vegetation into the sea, and that initially the Ws have a low proportion of ML, but that over time the proportion of ML increases (because the plastic litter persists while natural vegetation disintegrates and sinks). I understand that this question is not the goal of this study, but the proportion of Ws that have ML and therefore are classified as LWs does seem highly relevant and should be explored and presented if at all possible.

In general, the text sometimes is very difficult to follow, which is also due to the fact that the authors use a lot of abbreviations, not all of which are intuitive. I would advise to use fewer abbreviations and only focus on the most important and most frequently used ones. Similarly, concentrating the main text on the key results (figures 1 – 7) will substantially improve the manuscript.

1) Title: ok

2) Abstract: ok

3) Introduction: ok, motivates the present study well.

5) Methods: it would be good to add a conceptual figure in the manuscript that illustrates the key steps in this new approach; the figures in the supplement are very extensive and detailed, and a synthetic overview figure could be useful (while maintaining the detailed figures in the supplement).

6) Results & Discussion: the initial results are very interesting and nicely illustrate the stepwise approach and findings, but starting with figure 8, it seems that not all that info is necessary. In fact, starting with the subchapter “The case of the North Adriatic”, it seems that information becomes too detailed and some aspects are repetitive. It seems that the Adriatic subchapter, if it were to be maintained, should be moved to the electronic supplement. The subchapters about the optimal mission concept and then the scenario for research and management go beyond the principal goals of this study and should actually be drastically condensed and converted into to a very short and concise conclusion chapter. The current length of this results & discussion section substantially dilutes the overall impact of this new approach.

7) References: I have not checked whether all references are cited in the text and viceversa, but I feel that the authors could possibly draw a bit more on the Sargassum studies, which have followed a somewhat similar approach, for example (and several other papers): Dierssen, H.M., Chlus, A. and

Russell, B., 2015. Hyperspectral discrimination of floating mats of seagrass wrack and the macroalgae Sargassum in coastal waters of Greater Florida Bay using airborne remote sensing. *Remote Sensing of Environment*, 167, pp.247-258.

In summary, I think that this could become an important contribution, but it will require major revision and careful focusing of the text. This will likely mean a lot of additional work but I think that it is mainly careful crafting of the text, and so it should be doable. I encourage the authors to take on this task, because it will ensure the future impact of their paper and approach.

In case of any questions about my comments, I invite the authors to contact me directly.
Sincerely, Martin Thiel

Reviewer #2 (Remarks to the Author):

The paper reports on a very wide study aimed at defining specifications for an earth observation satellite mission with the purpose of detecting plastic litter at sea.

After motivating the use of a proxy (litter windrows) for detections, rather than aiming at the direct detection of scattered plastic items, the authors study a very large dataset covering the whole Mediterranean Sea for a long period of time, taken from Sentinel data, to make a convincing feasibility study for the mission.

Based on simulation of the reflectance properties of water, and of natural and artificial materials (specifically, the plastic polymers that are most represented in marine litter), at reasonable concentrations for real-world scenarios, an evaluation of the SNR obtained at satellite vs. the wavelength of received radiation (from UV to SWIR) is obtained. The most relevant bands according to the SNR for plastics, as well as for confounding materials, are chosen, discussing in this way the specification of a "super-spectral" sensor system for the mission envisioned.

This work is very relevant to a task that has been receiving considerable consideration recently, because of the impact of plastic litter on the marine environment. The results of this paper will contribute significantly to the design of a future international mission that may also tackle other pollution issues at sea.

The analysis of the literature in the field and of the state of the art is complete and relevant.

The work is described in a clear, complete, and reproducible way. Data analysis, interpretation, and conclusions appear conducted with a sound methodology.

The balance between the main paper and the large supplementary materials documents is in general appropriate, but the paper is not fully self-contained unless the supplementary material is taken into account. The latter is very wide and the sections are not perfectly consequent to each other. This is understandable because this document is not meant to be structured as a paper, but rather as a container for data that are not meant to be read at the same level of depth by all readers. However,

more direct coordination of the paper and the supplementary material is desirable. All the supplementary material should be essentially summarized and referenced in the paper, and be completely functional to adding detail and in-depth discussion to the corresponding sections of the paper. This is not always the case, and such coordination might be improved to facilitate readability and retrieval of information.

A specific instance of this concern is related to the choice of the bands indicated for the mission. I think that this issue, which is very important and central to the scope of the paper, is treated in the paper too rapidly, without an explanation of the methodology used for the choice of the bands and the bandwidth of each, which is to be found in the supplementary materials in at least two different sections, and with arguments not completely developed.

Another issue that I would like to highlight concerns the choice of Litter Windrows as a proxy for litter concentration. This choice is clearly and convincingly discussed in the paper and it is understandable that validation of the relation of LWD to the actual concentration of litter is very difficult due to the lack of sufficient ground truth. However, I think that this issue of validation should be discussed at greater length, and the feasibility of obtaining relevant ground truth assessed. Even if the correlation of LWD and actual litter density is very reasonable, it might be argued that different wind and currents conditions might cause LWs to build up differently (or not at all) even with the same litter quantities.

Lastly, I would ask the authors to comment in the paper on the possible use of data from the PRISMA mission to enhance the significance of their study. PRISMA hyperspectral data might be an important source of validation of the choice of the bands and bandwidths as discussed in this paper.

reviewed by Marco Balsi

POINT-BY-POINT REPLY FOR THE REVIEW

REVIEWER #1 (Remarks to the Author):

General evaluation: This manuscript presents interesting and potentially valuable results based on a new approach of detecting windrows from space, and confirming via spectral profiling that these windrows contain floating anthropogenic litter, in particular plastics. The general findings appear reasonable, namely more windrows in coastal regions, more litter windrows (LW) close to coastal sources and more abundant after rainfalls. Also the comparison of LWs with Plastic Inputs (PIs) appears suitable and generates very useful results (it basically is the Proof of Concept). I congratulate the authors for the idea and also for the immense work with image processing and the data analysis.

We thank the Reviewer for such a positive assessment and for acknowledging the dedication and effort that the group of authors has put into this work.

While the key results (figures 1-7) are impressive and relatively easy to follow, the structure of the manuscript is confusing and especially the concepts presented in the text require revision and focusing. The authors present a number of constructs (e.g. the "PoC detection processor") that are confusing in their logical conception and presentation. For example, they introduce the "Proof of Concept", which certainly is a very useful and valid approach (and the PoC itself is not the problem). In the present context, the PoC could mean that they propose a stepwise approach that passes through the remote detection of windrows, then the spectral profiling of plastic litter (which identifies among all the detected windrows those that indeed contain large quantities of plastic litter and results in the LWs), and via quantification of size (length and area) of these LW, they then determine the LWD (in ppm, which seems to be pretty much "m² per km²"). In a next step they then correlate these LWDs with estimated Plastic litter Inputs (PI), which is based on input source (=rivers?), runoff, and population density. They find a reasonably good correlation between LWD and PI, which, of course, is not perfect, but they identify the areas where important deviations from the overall correlation are found, and they provide reasonable arguments to explain these deviations (more local processes that cause these deviations). All this together (correlation between LWD & PI; with explanation of major deviations) then could be considered as a first "Proof of Concept" for this new approach. This seems to be the underlying "concept" (sic) of this study, and the main results follow this line of argumentation and testing of the PoC.

However, the authors then create substantial confusion by introducing the "PoC detection processor", which is a very odd construct. The PoC is here apparently converted to a specific "detector" that uses different spectral approaches and computing analysis steps to identify individual LWs. This (detection of Ws and identification of LW) is certainly an important part of this approach, but the linguistic construct of a "PoC detection processor" is very confusing and I strongly recommend to find a different construct for this, because the PoC is the umbrella concept that embeds the entire study: if the authors can show that their approach provides decent results that are useful to identify areas with high amounts of PL (accumulated in LW), then this is a reasonable Proof of Concept for their approach. In your analytical approach you certainly can have a detection processor, but I strongly recommend to give this a different identify than "PoC detection processor".

The Reviewer's comments are entirely correct. We recognise that the sequence of steps described in the previous version was not the most appropriate. The line of argumentation did not follow the sequence of steps in our work, and this fact indeed can lead to misunderstanding.

The temporal sequence of tasks in our work was not: remote detection of windrows => quantification of LW and LWD => relation with environmental/anthropogenic drivers => proposal of optimal sensor (as described in the initial version). The work started in 2017 with the design of an optimal mission concept, based on the spectral profiling of marine plastic debris. It was later (2021) when we rolled out the proof-of-concept to test the feasibility and usefulness of the mission concept by implementing it with best available technology on a real-world case study. This was done after realizing that Sentinel-2 had a partial compatibility with the required ideal instrument. With this PoC, we aimed to know whether LW abundance is sufficient to capture spatio-temporal patterns at any scale, explore the meaning of the LW proxy, and evaluate potential usefulness for addressing marine litter pollution.

ACTIONS:

1- The new version presents a more logical arrangement of the work. Now, it starts with the definition of the optimal mission concept (previously placed in a final section). Then we describe how, from this optimal concept, we derive a suboptimal approximation based on the in-orbit Sentinel-2.

2- We have included a simplified version of the former Fig. S1 into the main body (as suggested below). We believe that the inclusion of this figure at the beginning of the manuscript (Fig. 1 in the new version) is an important aid for the comprehensive understanding of the stepwise approach. Below, we show this new figure:

Fig. 1 | Roadmap followed for the definition of the optimal mission concept (left) and for the proof of concept (right). “Min. coverage” refers to the minimum fraction of plastic surface coverage required to identify a pixel with a true positive, estimated for both optimal mission and the proof of concept. An extended version of the workflow can be found in Fig. S1.

3- The construct "PoC detection processor" has been removed in the new version of the manuscript, as suggested by the Reviewer. "PoC detection processor" was replaced by "detector".

Another issue is the fact that the current approach only identifies LWs even though it seems that the authors might also be able to identify Windrows (Ws). I have to admit that I did not read the entire 137 pages of the electronic supplement, but I glanced through it and looked at different sections in order to better understand the overall approach. On P. 29 of the supplement, the authors say "To ensure our focus on LWs, we combine the spectral index with a contextual filter based on the geometry of such structures (section 2.1.8), and a further filtering was applied during post-processing (sections 2.2.2 and 2.3.3) to ensure that only LWs with high ML density were preserved for the PoC analysis." In the context of this study, it seems very important to understand where windrows are and which of these windrows really contain ML, and maybe even how much ML. As discussed in some of my comments in the annotated manuscript, I suspect that the rainfalls flush a lot of natural vegetation into the sea, and that initially the Ws have a low proportion of ML, but that over time the proportion of ML increases (because the plastic litter persists while natural vegetation disintegrates and sinks). I understand that this question is not the goal of this study, but the proportion of Ws that have ML and therefore are classified as LWs does seem highly relevant and should be explored and presented if at all possible.

Our detector is able to identify pixels with plastic-alike spectra (using the WSI spectral index, SI Section 2.1.7). Along with the windrows, we can identify patches with plastic-alike pixels and sections of the windrow with no apparent presence of plastic (see Fig. 2a). To generate a positive detection of a litter windrow, the pixels matching the plastic-alike spectra must make up a filament of considerable length (contextual filter). However, we did not keep those windrows that are not positive with the spectral index WSI that we designed. Many of them are not even detected with WSI, but we see the relevance of what the Reviewer proposes. This information may be of great interest to biological oceanographers and other researchers. Windrows without litter were not our goal, and unfortunately, they were not counted here. We take note of this interesting idea and will try to include this functionality in the next development of the detector and for new implementations.

We acknowledge that the Supplementary Information (SI) is extensive, and that some sections are quite technical. However, we believe it was necessary to include this extensive document to give the possibility to replicate and improve our advances. Below, we pick out some points from SI that refer to the comments raised by the Reviewer.

Regarding the Reviewer's question on the content of the windrows and potential quantification of marine litter, this matter is a conundrum for the Earth Observation community yet. The main reason why we need a dedicated satellite mission relies on various factors, being the lack of discrimination capability the main one. Up to date, very few existing missions have spectral resolution enough to theoretically describe floating plastic and separate materials within a windrow pixel. In particular, hyperspectral missions should have a much better chance, but their worse spatial resolution means they have to do better in terms of signal-to-noise to achieve detection. A reduced number of pixels makes it much harder to differentiate a windrow from

noise. The noise can be caused by instrument artifacts, observing conditions like clouds, glint, or closeness to the edges of the range of detection. Our technological assessment during the definition of the optimal mission concept, led us to conclude that the threshold value of spatial resolution to describe properly a windrow is 10 m/pixel, and ideally, we should try to get the value down up to 5m or even 3m (current hyperspectral missions are in 30m/pixel). The challenge, however, is to get sufficient signal-to-noise at these resolutions, the reason why 10m is safer and closer to what can be done with current technologies.

With that in mind, from all the flying non-commercial missions, only Sentinel-2 meets the requirements of spatial resolution, and several of its bands are compatible with those derived in the mission concept. It is not perfect though (otherwise we would not need a specific mission), and the number of compatible bands is too low for a complete discrimination. Additionally, Sentinel-2 has an uneven spatial sampling that complicates things. The work from Hu (2022, ref. 21) is a good read to properly understand the challenges and limitations of Sentinel-2 for marine litter, but in a nutshell, Hu (2022) concludes that the possibility to separate marine litter from other non-photosynthetically active materials with this satellite is quite limited. This does not mean it is not possible, but so far, none of the published papers on the topic has presented a solid methodology for the spectral unmixing required to do this distinction within the non-photosynthetic floating material fraction.

The reason this spectral unmixing is necessary is that marine litter submesoscale accumulations of floating matter will rarely be composed exclusively of marine litter or plastics. In most cases, we will be talking about a mixture of things. Accompanying materials are often driftwood, leaves and the likes, particularly after rainfall, as indicated correctly by the Reviewer. Another common accompanying matter is metastable seafoam. The accompanying materials impact the apparent color of the pixel (land-based materials tend to be brown; seafoam tends to be white/yellowish), which means relying on “color” for separation is prone to errors. It is in the NIR and SWIR bands where we have more discrimination options for marine litter, being this one of the main reasons driving the design of the WSI index. We, indeed, explored this issue in our study.

In SI Section 2.1.7, we introduce the “Normalized Spectral Index Confusion Matrix”. The idea behind this method is to study relationships between the spectral bands of a sensor for a set of target materials (or pixel classes) and help to identify which pairs of bands contain the maximum information about one spectral class, and how those pairs of bands behave in other materials. Only the pairs that maximize information and reduce the confusion between materials shall be selected. This is how we isolated the target bands in Sentinel-2, which were then compared with those identified for the ideal sensor.

When computing Normalized Spectral Index Confusion Matrix using selected patches of Sentinel-2 pixels for different spectral classes, marine litter provides a specific signature vs. the other classes, with some specific bands being better than others. We explored those bands to create our WSI spectral index. This method does not guarantee a 100% successful distinction, but allows us to choose the appropriate bands to maximize the chances that a detection corresponds to a given pixel class and not to another. The thresholding method applied to WSI obeys the need to establish a differentiation between what is considered as marine litter and what is not.

Increasing ground-truth information also helps in the process. In Section 2.4 of the Supplementary material, we introduced some validation tests of this method, using observed LWs with visual confirmation from local observers (MARIDA database includes some of those), along with a technique to convert WSI values into marine litter fractions. The results of this method are presented in an example in the Supplementary Figure 9. There, readers can see how it is possible to calibrate WSI to produce an estimation of fraction. However, this method is not a spectral separation or unmixing, but rather an estimation of the amount of material falling within the general marine litter class present in a pixel, taking into account that this class embeds not only litter, but also some of the accompanying materials (seaweeds, driftwood, seafoam) in unknown proportions. Because the proposed method is context-dependent and has to be calibrated for each windrow, it is not possible to train it using some well-known LWs to estimate the composition of other LWs detected by the satellite. Thus, we get limited in our results to report the positive pixels. We used this method on the well-known LWs to identify the lower limit detection threshold for Sentinel-2 sensor (20%). Future work may go in this direction, in order to refine the results and provide further information.

Thus, the combination of WSI with a threshold system is what allows our method to isolate pixels where there is most likely to be marine litter and in particular plastic, due to the use of plastic spectral signatures in the definition of the WSI index. But for now, we cannot quantify the exact composition of such marine litter. The post-filtering layers we added and the manual screening we also did are intended to remove filaments that are not what we are looking for. In particular, the use of NDVI and NDWI in the post-filtering (SI Section 2.2.2) not only removes “artifacts”, but also vegetation-like, or photosynthetically active filaments by construction of the filter ranges. This means that we only retain windrows that are mostly composed of non-photosynthetic materials.

ACTIONS:

1. Our work advances the spectral detection method of plastic debris by using, for the first time, an infrared-based spectral index (WSI) and other methods to remove artifacts (e.g. windrow composed of vegetation-like material). In addition, we found a reasonably good correlation of LWD with plastic inputs and basin-scale data. However, we agree that we must make it clear that our detections should be understood as aggregations containing varying fractions of anthropogenic and natural debris. This limitation arises from the technical constraints of the suboptimal sensor used (S2-MSI), and it is common to all papers dealing with plastic detection through this sensor, although this has not always been made clear. In the new version of the manuscript, we have removed any doubt that might lead readers to believe that our detections correspond unequivocally to plastic. The “Optimal concept and proof of concept” section now reads (new text is underlined):

“While the spectral detection was targeted on plastic, detections must be considered here as ML aggregations containing varying fractions of both anthropogenic and natural debris, due to the known limitations of the S2-MSI²¹. Eventually, with dedicated spectral bands (Table 1), it will be possible to disentangle the contributions of the main litter components (e.g., plastic, weeds, driftwood, metastable seafoam) to the remotely-sensed reflectance and estimate plastic concentrations per pixel.”

Please, note that we REMOVED the following sentences from this paragraph (as suggested in the annotated version of the revised manuscript):

“Nevertheless, in the Mediterranean Sea, the majority of all floating items are plastics (> 75%, accounting for both anthropogenic and natural) and the abundance of natural debris has been found to correlate positively with plastic concentrations^{22,23}. Therefore, the ROI attributes led us to expect a potentially significant relationship between our PoC detections and plastic dynamics”.

2. Following the suggestions of Reviewer #2, we have included a new paragraph in the last section of the manuscript describing how to build on the progress of this work to increase the collection of field data on LWs, and thus advance the determination of the contributions of their main litter components. It is a Reviewer #2's suggestion but we believe it fits well with Reviewer #1's commentary as well. This new paragraph reads:

“Next steps should also advance on the ground truthing (Fig. 1). The current matching of satellite detections and field observations is limited to fake targets (artificial LWs)^{15,20} and reports of dense LW sightings¹⁴ (Figs. S9 and S11). Filling this gap requires targeted and coordinated research on the LW phenomenon and the combination of several approaches. Fake targets are an effective approach, which might be further extended to different LW compositions and environmental conditions. Matching detections of natural LWs and field measurements is a more challenging task due to the difficulty of finding LWs from ship surveys¹³. We can now take advantage of satellites to address areas and periods of common LW formation (Fig. 5). Complementary observation platforms with larger coverage area than ships (e.g. long-range drones, aircrafts) could also significantly increase LW findings²⁷. Additional inputs for calibration and validation can be gained from the ocean-wide spreading of animal-borne cameras⁴⁷, particularly from species using windrows as feeding spots. Citizen science could also contribute with opportunistic observations, whereas from leisure vessels, fishing, or commercial ships. Finally, another pivotal approach stems from the overlapping of S2-MSI detections with observations from medium-to-high resolution hyperspectral satellites⁴⁸.”

3. We have improved the linkage between the main manuscript and the key points of Supplementary Information, in order to facilitate the use of this material for more specific questions.

In general, the text sometimes is very difficult to follow, which is also due to the fact that the authors use a lot of abbreviations, not all of which are intuitive. I would advise to use fewer abbreviations and only focus on the most important and most frequently used ones. Similarly, concentrating the main text on the key results (figures 1 – 7) will substantially improve the manuscript.

Again, the Reviewer is right in his comment. The work was quite long and dense, and indeed a simplification of the text together with a shortening have significantly contributed to its improvement.

ACTIONS:

1. We have reduced the total number of abbreviations by 20%, focusing on the most important and most frequently used abbreviations. This reduction of abbreviations has been even more intense in the main manuscript (half of the abbreviations (11) have been deleted in the revised version of the main manuscript). As suggested in the annotated manuscript, we have also written out the acronyms in the figure captions.

2. We have considerably reduced the text of the main manuscript, moving the section on the Northern Adriatic and the exploration of possible applications to the SI (as suggested by the Reviewer in comments below).

3. We have reduced the number of figures. In the new version, figures 1 to 7 remain (as suggested by the Reviewer), and figure 9 was moved to Supplementary Information. Figure 8 was a variant of the current figure 4, so we have removed it and included a simplified version in an inset in figure 4. These changes have resulted in a more synthesized, orderly and logical presentation of the figures. The new figure 4 (with “b” inset) is shown below:

1) Title: ok

2) Abstract: ok

3) Introduction: ok, motivates the present study well.

5) Methods: it would be good to add a conceptual figure in the manuscript that illustrates the key steps in this new approach; the figures in the supplement are very extensive and detailed, and a synthetic overview figure could be useful (while maintaining the detailed figures in the supplement).

Indeed, given the magnitude of the work, an initial figure providing a synthetic overview is necessary for an easier understanding of the work. Thanks for this observation.

ACTION: The SI figure with the synthetic overview referred to by the Reviewer is the previous Fig. S1. This figure remains in SI, but a simplified version is now shown in the revised version of the main manuscript. This new figure is shown in a reply above (new Fig. 1).

6) Results & Discussion: the initial results are very interesting and nicely illustrate the stepwise approach and findings, but starting with figure 8, it seems that not all that info is necessary. In fact, starting with the subchapter "The case of the North Adriatic", it seems that information becomes too detailed and some aspects are repetitive. It seems that the Adriatic subchapter, if it were to be maintained, should be moved to the electronic supplement. The subchapters about the optimal mission concept and then the scenario for research and management go beyond the principal goals of this study and should actually be drastically condensed and converted into a very short and concise conclusion chapter. The current length of this results & discussion section substantially dilutes the overall impact of this new approach.

As commented above, we have kept the stepwise presentation up to figure 7, and moved to SI the sections and figures indicated by the Reviewer. These changes have resulted in a shortening of the main paper to focus on the most relevant findings.

ACTIONS:

1. Figure 9 was moved to Supplementary Information, as suggested. We considered that figure 8 was necessary for a better understanding. So, we have included a simplified version in an inset in figure 4. The new figure 4 is shown in the responses above.

2. The section about the "optimal mission concept", once simplified and merged with the former "PoC detection of marine litter windrows" section, has been moved to the beginning of the manuscript. The section resulting from this merging is entitled: "Optimal concept and proof of concept". This initial section now provides a synthetic overview of the work in a more logical order, which we believe facilitates the understanding of the reasoning behind the work.

3. The section "The case of the North Adriatic" has been moved to SI (Section S4), as suggested.

4. The text regarding the exploration of practical application cases, in the section "A new scenario for research and management", has also been moved to SI, as suggested. As a result, this section has been shortened and simplified. The long paragraph dealing with the exploration of application cases has been reduced to a sentence linking to SI. This sentence reads:

"Using real case scenarios, Supplementary Information illustrates some applications, such as the identification of pollution hotspots, shipping-related sources of litter, long-term evaluation of action plans or better targeting of clean-up efforts (SI Section 4, Fig. S33)."

7) References: I have not checked whether all references are cited in the text and vice versa, but I feel that the authors could possibly draw a bit more on the Sargassum studies, which have

followed a somewhat similar approach, for example (and several other papers): Dierssen, H.M., Chlus, A. and Russell, B., 2015. Hyperspectral discrimination of floating mats of seagrass wrack and the macroalgae Sargassum in coastal waters of Greater Florida Bay using airborne remote sensing. Remote Sensing of Environment, 167, pp.247-258.

We have included the references suggested by both Reviewers, and we have checked the references cited in the text and vice versa. We thank the Reviewers for their suggestions of articles, especially the relevant article by Tramoy et al. (suggested in the annotated manuscript), which provides field evidence of the strong input of litter recorded by satellite in southern France during the spring of 2018. On the other hand, the shortening of the paper has resulted in a transfer of references from the main manuscript to the SI.

ACTION: The following references have been added to the revised version of the main manuscript:

- Tramoy, R., Blin, E., Poitou, I., Noûs, C., Tassin, B. and Gasperi, J., 2022. Riverine litter in a small urban river in Marseille, France: Plastic load and management challenges. *Waste Manag.* **140**, 154-163 (2022).
- Laverre, M. et al. Heavy rains control the floating macroplastic inputs into the sea from coastal Mediterranean rivers: A case study on the Têt River (NW Mediterranean Sea). *Sci. Total Environ.* **877**, 162733 (2023).
- Korshenko, E., Zhurbas, V., Osadchiv, A. & Belyakova, P. Fate of river-borne floating litter during the flooding event in the northeastern part of the Black Sea in October 2018. *Mar Pollut Bull.* **160**: 111678 (2020)
- Nishizawa, B., Thiebot, JB., Sato, F. et al. Mapping marine debris encountered by albatrosses tracked over oceanic waters. *Sci. Rep.* **11**, 10944 (2021).
- Corbari, L., Capodici, F., Ciraolo, G., & Topouzelis, K. Marine plastic detection using PRISMA hyperspectral satellite imagery in a controlled environment. *Int. J. Remote Sens.* **44**, 6845-6859 (2023).

The following reference have been added to the new version of SI (as suggested by the Reviewer):

- Dierssen, H.M., Chlus, A., & Russell, B. Hyperspectral discrimination of floating mats of seagrass wrack and the macroalgae Sargassum in coastal waters of Greater Florida Bay using airborne remote sensing. *Remote Sens. Environ.* **167**, 247-258 (2015).

In summary, I think that this could become an important contribution, but it will require major revision and careful focusing of the text. This will likely mean a lot of additional work but I think that it is mainly careful crafting of the text, and so it should be doable. I encourage the authors to take on this task, because it will ensure the future impact of their paper and approach.

We are sincerely very satisfied with the changes suggested by the Reviewers. Thanks for the contribution.

In case of any questions about my comments, I invite the authors to contact me directly.

Sincerely, Martin Thiel

Facultad de Ciencias del Mar, Universidad Católica del Norte, Larrondo 1281, Coquimbo, Chile

We thank Prof. Martin Thiel for his helpful suggestions and availability.

Other minor ACTIONS in response to the suggestions included in the annotated version of the manuscript included:

1. In the new version of the main manuscript, we have written the years (instead of “January Year”) on x-axis in the figures (Figs. 7 and 8 in the revised version of the manuscript).

2. In the new version of “Introduction”, we have clarified that windrows can also be referred to as "filaments" (as suggested in the annotated version of the revised manuscript). Introduction now reads (new text underlined):

“...the metre-sized aggregations of floating debris, the so-called surface slicks, streaks, filaments or litter windrows”

REVIEWER #2 (Remarks to the Author):

The paper reports on a very wide study aimed at defining specifications for an earth observation satellite mission with the purpose of detecting plastic litter at sea. After motivating the use of a proxy (litter windrows) for detections, rather than aiming at the direct detection of scattered plastic items, the authors study a very large dataset covering the whole Mediterranean Sea for a long period of time, taken from Sentinel data, to make a convincing feasibility study for the mission. Based on simulation of the reflectance properties of water, and of natural and artificial materials (specifically, the plastic polymers that are most represented in marine litter), at reasonable concentrations for real-world scenarios, an evaluation of the SNR obtained at satellite vs. the wavelength of received radiation (from UV to SWIR) is obtained. The most relevant bands according to the SNR for plastics, as well as for confounding materials, are chosen, discussing in this way the specification of a "super-spectral" sensor system for the mission envisioned.

This work is very relevant to a task that has been receiving considerable consideration recently, because of the impact of plastic litter on the marine environment. The results of this paper will contribute significantly to the design of a future international mission that may also tackle other pollution issues at sea.

We are grateful to Reviewer #2 for the thorough reading and appreciation of our work, as well as for the comments provided, all of which were very helpful in improving the manuscript. The scope of the manuscript is broad, so we are happy that the main ideas we wanted to convey are well captured by an external reader.

The analysis of the literature in the field and of the state of the art is complete and relevant.

We thank the Reviewer for checking the provided references and the assessment on the state of the art.

The work is described in a clear, complete, and reproducible way. Data analysis, interpretation, and conclusions appear conducted with a sound methodology.

We appreciate such positive comments.

The balance between the main paper and the large supplementary materials documents is in general appropriate, but the paper is not fully self-contained unless the supplementary material is taken into account. The latter is very wide and the sections are not perfectly consequent to each other. This is understandable because this document is not meant to be structured as a paper, but rather as a container for data that are not meant to be read at the same level of depth by all readers. However, more direct coordination of the paper and the supplementary material is desirable. All the supplementary material should be essentially summarized and referenced in the paper, and be completely functional to adding detail and in-depth discussion to the corresponding sections of the paper. This is not always the case, and such coordination might be improved to facilitate readability and retrieval of information.

We thank the Reviewer for going through the extensive Supplemental Information. The present manuscript is the result of several years of work, so for the sake of reproducibility and transparency, we were obliged to present much of the methods, experiments and analyses in an extensive and detailed supplementary document.

The Reviewer is right: we intended the document to be consultative and not to be read systematically, putting it in the order that the different parts of the study were done, which is why sometimes it may not look consequential. However, only in this way can one follow the decision-making process and how we converged towards the final approach.

We agree on the importance of improving the linkage between the main paper and the sections of the Supplemental Information, so readers can be guided to the right sections if they wish for details on the methods or further discussion.

ACTIONS:

1. Based on the comments of Reviewer #1, we have changed the order in which ideas are presented in the manuscript. The revised version of the main manuscript now starts with the definition of the optimal mission concept and then we describe how we derive a suboptimal approximation based on the in-orbit Sentinel-2. This change has resulted in a more logical arrangement of the work, adjusted to the real-time sequence of steps followed in our work. Importantly, now the revised version of the manuscript follows a structure parallel to that in SI, resulting in an easier linkage between the two documents.

2. We have revised throughout the manuscript the linkage to the Supplementary Information, which we believe has now been considerably improved (referencing highlighted in yellow in the revised version of the manuscript with tracked changes). To avoid impeding a smooth reading of the main text, though, we have tried to find a compromise between the number of times a cross-reference was introduced in the text and an easier reading. We hope the additions made in this regard are found satisfactory.

A specific instance of this concern is related to the choice of the bands indicated for the mission. I think that this issue, which is very important and central to the scope of the paper, is treated in the paper too rapidly, without an explanation of the methodology used for the choice of the bands and the bandwidth of each, which is to be found in the supplementary materials in at least two different sections, and with arguments not completely developed.

We agree that we have addressed this issue too succinctly in the main paper. The reason was the space constraints as well as that we decided to put the main focus on the proof of concept. However, it is true that the optimal mission concept is also very important and more attention should be given. As Referee #1 asked for some text reshuffling and cleaning on the main manuscript, we have taken the chance to introduce in the main manuscript some additional information on the band selection process and also a better link to the relevant sections of the Supplemental Material.

ACTIONS:

1. As commented above, following Reviewer #1 comments, the definition of the optimal mission now has a more prominent place in the manuscript, being now presented in the first section of

the revised manuscript. Also, the new figure 1 presents a comprehensive framework of the work, showing both the definition of the optimal concept and the proof of concept, and their interrelationship. This new figure is shown above in the responses to Reviewer #1

2. Although we tried not to increase the length of the manuscript (see Reviewer #1 comments), we have added new text and links to SI sections regarding band selection. The revised version of the manuscript now reads (new text underlined):

“...This spectral library was fed into an atmospheric radiative transfer model in order to determine spectral radiances at the top of the atmosphere. The isolation of spectral bands for the sensor, and their width, was achieved by optimising the remote sensing performance (as signal-to-noise ratio) with changing fractions of plastic polymers over a range of observation conditions (SI Sections S1.2.2 and S3.1.2, Figs. S13 to S21). A set of additional bands was configured to specifically address the quantification of other abundant floating materials (i.e. algae, driftwood and seafoam) and to support atmospheric correction and cloud detection (Table S3). This exercise allowed us to define functionalities for a total of 23 candidate spectral bands (Table 1, Figs. S22 to S24).”

Another issue that I would like to highlight concerns the choice of Litter Windrows as a proxy for litter concentration. This choice is clearly and convincingly discussed in the paper and it is understandable that validation of the relation of LWD to the actual concentration of litter is very difficult due to the lack of sufficient ground truth. However, I think that this issue of validation should be discussed at greater length, and the feasibility of obtaining relevant ground truth assessed. Even if the correlation of LWD and actual litter density is very reasonable, it might be argued that different wind and currents conditions might cause LWs to build up differently (or not at all) even with the same litter quantities.

We fully agree with the Reviewer that field validation is a weak point for satellite remote sensing of marine litter. The problem does not impact solely this study, but overall, all published studies with pragmatic approaches to existing data suffer from the same limitations. A large part of the reason why we conducted the Proof of Concept lies precisely in this argument: lacking enough ground truth for a solid direct validation, focusing on the geostatistical significance of a large data set of LW detections, and relating it to existing knowledge on marine litter pollution, seemed like the only possible way to proceed.

The results were highly consistent with such knowledge and with the main environmental drivers involved in marine litter distribution. Nevertheless, we agree with the Reviewer that improving ground truthing is an important next step, which in turn can benefit from the progress made with our manuscript. The ephemeral nature of the windrows and their relatively reduced spatial scales make direct observation at sea very difficult. The revisiting time of many satellites, even if in Sentinel-2 is as short as 5 days on average, is not sufficient for tracking detected structures nor facilitates having match-ups with field observations to perform direct validations.

Our manuscript now allows us to take advantage of satellites to address areas and periods of common LW formation. In addition, we need targeted and coordinated research on the LW phenomenon as well as the combination of multiple approaches to increase ground truthing.

Setting out in the manuscript how this next step could be carried out seems to us a very wise suggestion, and we have done so in the new version of the manuscript (see actions below).

Possible strategies that should be put in place include:

1. Use of fake targets, as done in Lesbos Island (Greece)
2. Further coordination and international collaboration for field sampling, not only by means of ships but also by other means with larger coverage, such as airplanes, helicopters, and long-range drones, probably all of them supported by the information that can be generated by Sentinel-2. Some initiatives, such as the Ocean Scan (<https://www.oceanscan.org/>) and the IOCCG Taskforce for Remote Sensing of Marine Litter and Debris (<https://ioccg.org/group/marine-litter-debris>) could maybe foster these actions.
3. Use other missions to better study the detections done by Sentinel-2 (e.g., PRISMA, should concomitant data at the right time be found).
4. Further methodology testing using hyperspectral sensors mounted on aircrafts.
5. Citizen science to support proper reporting of identified litter windrows at sea (e.g., further involvement of fishermen and environmental policing agencies)

Regarding the building of LWs, in the main manuscript, we have specifically accounted for the role of wind in order to obtain more realistic estimates of litter pollution:

“Quantitative assessment of ML pollution on a continuous scale was done through the fraction of sea surface area covered by LWs, referred to as litter-windrow density (LWD, in ppm; e.g., m^2 per km^2). Importantly, LWD was calculated solely over the sea surface under environmental conditions deemed suitable for litter aggregation, i.e., for winds lower than $5 m s^{-1}$ (see Methods). By reducing spatio-temporal bias due to different wind conditions and, consequently, varying conditions for LW formation, this wind-based constraint was intended to improve the comparability of LW detections in space and time, an issue hitherto unaccounted for”.

In SI figure S27, we show the apparent role of wind speed on LWs, which seem to break off when winds are above 5 m/s.

Supplementary Information regarding the consideration of wind on LW detections is now more suitably referenced in the revised version of the main manuscript. Moreover, we agree that the effect of physical forcing on LW formation must be addressed in future research, as this has not yet been fully disclosed by our research. This idea is now more clearly visible in the last section of the revised manuscript.

In short, we have made some small changes to the revised version of the manuscript to stress the difficulty of direct validation and how we might address this gap, as well as to better frame the range of validity of the work undertaken.

ACTIONS:

1. On the need to move forward on the physical control of LW formation, we now have a single paragraph in the final section that addresses this issue. This single paragraph reads as follows:

“The readiness level of the present monitoring tool allows for its implementation in other ocean regions; however, there is still much room for improvement. S2-MSI surveys in other ocean regions could advance the mission concept, analysis methods and data interpretation. Current LWD estimates account for wind conditions, since LW formation requires not only high ML load, but also favourable sea state¹³. Nevertheless, further refinement of LWD estimates should ideally account, along with disruptive forces (e.g. wind mixing), for major formation forces (e.g. tidal forcing).”

2. Regarding the ground truthing and how we might address this limitation, a specific paragraph was introduced in the new version of the manuscript. This new paragraph reads:

“Next steps should also advance on the ground truthing (Fig. 1). The current matching of satellite detections and field observations is limited to fake targets (artificial LWs)^{15,20} and reports of dense LW sightings¹⁴ (Figs. S9 and S11). Filling this gap requires targeted and coordinated research on the LW phenomenon and the combination of several approaches. Fake targets are an effective approach, which might be further extended to different LW compositions and environmental conditions. Matching detections of natural LWs and field measurements is a more challenging task due to the difficulty of finding LWs from ship surveys¹³. We can now take advantage of satellites to address areas and periods of common LW formation (Fig. 5). Complementary observation platforms with larger coverage area than ships (e.g. long-range drones, aircrafts) could also significantly increase LW findings²⁷. Additional inputs for calibration and validation can be gained from the ocean-wide spreading of animal-borne cameras⁴⁷, particularly from species using windrows as feeding spots. Citizen science could also contribute with opportunistic observations, whereas from leisure vessels, fishing, or commercial ships. Finally, another pivotal approach stems from the overlapping of S2-MSI detections with observations from medium-to-high resolution hyperspectral satellites⁴⁸.”

Lastly, I would ask the authors to comment in the paper on the possible use of data from the PRISMA mission to enhance the significance of their study. PRISMA hyperspectral data might be an important source of validation of the choice of the bands and bandwidths as discussed in this paper.

Indeed, the Reviewer is correct in indicating hyperspectral missions as having the potential to support the development of a dedicated mission. PRISMA, along with other hyperspectral missions such as EnMAP, have been considered during the technological feasibility phase of this study. In Section 3.2.2 of the Supplemental Information (SI), we referred to such missions and emphasized that “these missions offer opportunities for testing marine litter detection methods directly in the spectral bands of interest”. We also mentioned, in the same section, their potential added value to support the building and testing of end-to-end simulators. We studied PRISMA as part of our technology evaluation to find current sensors and data that may be used to report marine litter (SI Table S5). We did not mention the mission, though, in the main manuscript. We fully agree that, once the right bands have been identified, those could be exploited by existing hyperspectral sensors to test their potential for this task.

In terms of PRISMA capabilities for marine litter, we considered a publication from Corbari *et al.* (2023) titled: “*Marine plastic detection using PRISMA hyperspectral satellite imagery in a controlled environment.*”

The results of the paper are based on a limited set of collocations between PRISMA, PlanetScope, and drone spectra data over the artificial targets deployed on Lesbos Island (Greece) by the University of Aegean. Even under those controlled conditions, the authors found that PRISMA has some limited capability, which mainly worked for detection and not as much for quantification, an objective we endorse for a future dedicated mission. The authors did not manage to fully geolocate targets with PRISMA pixels (Fig. 5 of the publication), so validating the retrieved fractions vs. ground truth would be difficult. They also acknowledge that the PRISMA spatial resolution may be too coarse for this type of application. In contrast, they showed value in using spectral profiling derived from PRISMA for marine litter detection.

In the Supplementary Information, we demonstrate how the key aspect is the use of spectral anomalies, and not much the use of bulk reflectance measurements, understanding such anomalies as values with respect to seawater. The additional bands of a hyperspectral mission offer more opportunities for detecting anomalies (this is well illustrated in Corbari *et al.* 2023), but wider bands offer better anomaly detection. We focus on this aspect since spectral anomalies can be related to detection and quantification, and those can be better constrained when more bands are available (i.e., better spectral resolution). However, the compromise needed between spectral sampling and signal-to-noise ratio led us to conclude that an intermediate solution was needed. A multispectral sensor will have good bands but not in enough quantity. A hyperspectral sensor would have many bands but not so good (at least to the required spatial resolution). The super-spectral instrument falls in between and tries to get the best of both worlds as a compromise.

An optimal solution would require both spectral and spatial resolution, as well as sufficient signal-to-noise ratio to be fully functional. Anomalies can help improve SNR by removing unwanted information, so they work better for hyperspectral sensors, where spectral uncertainties may be higher than in instruments with wider bands. Decluttering satellite spectral data quite often involves reducing spatial resolution for most applications. This, in turn, forces a better signal-to-noise ratio in the averaged pixel to trigger the detection of a sub-pixel structure that would also be of smaller apparent size for the pixel size.

Consequently, the difficulty in applying PRISMA to validation lies in the challenge of establishing a clear linkage between ground truth data and PRISMA detections, so that the latter may be utilized to determine ideal bands or to test proposals. This issue is affecting most (if not all) current satellite data that does not meet the required signal-to-noise ratio, lacks the required bands, or misses the needed spatial resolution for the intended application. It is for those reasons that, generally, using an ASD FieldSpec instrument for field measurements along with radiative transfer modeling seems more adequate at this stage of mission concept development. This is the approach that we followed in our work. On the other hand, we agree that proper research should be done to test whether current satellite hyperspectral missions could help determine the feasibility of the proposed Mission Concept. For this, dedicated experiments are required to solve the issue of linking remote observations and ground truth data. This could be a good part of an end-to-end simulator exercise to further develop the mission idea in the future.

In this spirit, it is a great idea to mention such a possibility in the main manuscript and refer to PRISMA as an example of a mission that could fulfill such a role.

ACTION: We improved the linkage of the main body with the SI sections related to the possible use of hyperspectral data in the definition and validation of the optimal mission concept. Moreover, in the new paragraph included as action in response to the previous comment, we have specifically highlighted the option here discussed and proposed by Reviewer #2. This new sentence reads:

“Finally, another pivotal approach stems from the overlapping of S2-MSI detections with observations from medium-to-high resolution hyperspectral satellites⁴⁸.”

Reference number 48 in the new version of the manuscript corresponds specifically to PRISMA, as suggested, in particular to the following article:

- Corbari, L., Capodici, F., Ciraolo, G., & Topouzelis, K. Marine plastic detection using PRISMA hyperspectral satellite imagery in a controlled environment. *Int. J. Remote Sens.* **44**, 6845-6859 (2023).

REVIEWERS' COMMENTS

Reviewer #1 (Remarks to the Author):

great job in revising the manuscript. there are a few minor edits in the annotated manuscript. I would recommend to not talk about "game changer" because that is a very ambiguous term, and i do not think that your tool on its own will be a game changer. it is a useful tool that can be applied to many different tasks, but the game is being changed elsewhere, namely in the political arena!

best wishes, martin

Reviewer #2 (Remarks to the Author):

The paper reports on a very wide study aimed at defining specifications for an earth observation satellite mission with the purpose of detecting plastic litter at sea.

After motivating the use of a proxy (litter windrows) for detections, rather than aiming at the direct detection of scattered plastic items, the authors study a very large dataset covering the whole Mediterranean Sea for a long period of time, taken from Sentinel data, to make a convincing feasibility study for the mission.

Based on simulation of the reflectance properties of water, and of natural and artificial materials (specifically, the plastic polymers that are most represented in marine litter), at reasonable concentrations for real-world scenarios, an evaluation of the SNR obtained at satellite vs. the wavelength of received radiation (from UV to SWIR) is obtained. The most relevant bands according to the SNR for plastics, as well as for confounding materials, are chosen, discussing in this way the specification of a "super-spectral" sensor system for the mission envisioned.

This work is very relevant to a task that has been receiving considerable consideration recently, because of the impact of plastic litter on the marine environment. The results of this paper will contribute significantly to the design of a future international mission that may also tackle other pollution issues at sea.

The analysis of the literature in the field and of the state of the art is complete and relevant.

The work is described in a clear, complete, and reproducible way. A very large supplementary document contains details covering all aspects of the work. Data analysis, interpretation, and conclusions appear conducted with a sound methodology.

The authors reviewed the papers according to my concerns about the first version submitted, in a completely satisfying way.

Just a very minor note: please check the units for kinetic energy in figs. 3 and 4.

Subject: Nature Communications Ms. NCOMMS-23-59779 "Proof of concept for a new sensor to monitor marine litter from space". Reply to Reviewers.

Dear Reviewers,

Thank you for this new review of our manuscript. We are very grateful for your comments and suggestions. The review process has greatly improved the quality of the manuscript. In this new version, we have adjusted the manuscript to the style established by the Nature Communications guidelines.

We reproduce below the Reviewers' comments (in blue) and provide the corresponding response.

Reviewer #1 (Remarks to the Author):

Great job in revising the manuscript. There are a few minor edits in the annotated manuscript. I would recommend to not talk about "game changer" because that is a very ambiguous term, and I do not think that your tool on its own will be a game changer. it is a useful tool that can be applied to many different tasks, but the game is being changed elsewhere, namely in the politicalarena!

Response: We thank Reviewer #1 for the thorough review of our work. The comments provided throughout this review process have been of great help in improving our manuscript.

We have discussed carefully with the co-authors the suggestion to replace "game changer" with "tool". We have finally opted to keep the term "game changer". We fully believe that the incorporation of satellite technology into the "game" of marine debris monitoring can be regarded, based on the results shown in the manuscript, as a game changer.

Reviewer #2 (Remarks to the Author):

The paper reports on a very wide study aimed at defining specifications for an earth observation satellite mission with the purpose of detecting plastic litter at sea.

After motivating the use of a proxy (litter windrows) for detections, rather than aiming at the direct detection of scattered plastic items, the authors study a very large dataset covering the whole Mediterranean Sea for a long period of time, taken from Sentinel data, to make a convincing feasibility study for the mission.

Based on simulation of the reflectance properties of water, and of natural and artificial materials (specifically, the plastic polymers that are most represented in marine litter), at reasonable concentrations for real-world scenarios, an evaluation of the SNR obtained at satellite vs. the wavelength of received radiation (from UV to SWIR) is obtained. The most relevant bands according to the SNR for plastics, as well as for confounding materials, are

chosen, discussing in this way the specification of a "super-spectral" sensor system for the mission envisioned.

This work is very relevant to a task that has been receiving considerable consideration recently, because of the impact of plastic litter on the marine environment. The results of this paper will contribute significantly to the design of a future international mission that may also tackle other pollution issues at sea.

The analysis of the literature in the field and of the state of the art is complete and relevant. The work is described in a clear, complete, and reproducible way. A very large supplementary document contains details covering all aspects of the work. Data analysis, interpretation, and conclusions appear conducted with a sound methodology. The authors reviewed the papers according to my concerns about the first version submitted, in a completely satisfying way.

Just a very minor note: please check the units for kinetic energy in figs. 3 and 4.

Response: We thank the Reviewer for such a positive assessment.
We have checked the units for kinetic energy in figs. 3 and 4. They are correct.

We hope that you will consider this new version of the manuscript suitable for publication.

Yours sincerely,

Andrés Cózar, Manuel Arias and co-authors